# C. elegans SSNA-1 is required for the structural integrity of centrioles and bipolar spindle assembly

Jason A. Pfister [1,5], Lorenzo Agostini [2,5], Lorène Bournonville [3], Aurélien Perrier[3], Prabhu Sankaralingam [1], Zachary G. Bell [1], Virginie Hamel [3], Paul Guichard [3], Christian Biertümpfel [2], Naoko Mizuno [2,4] ✉ & Kevin F. O'Connell [1] ✉

Centrioles play key roles in mitotic spindle assembly. Once assembled, centrioles exhibit long-term stability, but how stability is achieved and how it is regulated are not completely understood. In this study we show that SSNA-1, the *Caenorhabditis elegans* ortholog of Sjogren's Syndrome Nuclear Antigen 1, is a constituent of centrioles and centriole satellite-like structures. A deletion of *ssna-1* results in the formation of extra centrioles. We show that SSNA-1 genetically interacts with the centriole stability factor SAS-1 and is required post assembly for centriole structural integrity. In SSNA-1's absence, centrioles assemble but fracture leading to extra spindle poles. However, if the efficiency of cartwheel assembly is reduced, the absence of SSNA-1 results in daughter centriole loss and monopolar spindles, indicating that the cartwheel and SSNA-1 cooperate to stabilize centrioles during assembly. Our work thus shows that SSNA-1 contributes to centriole stability during and after assembly, thereby ensuring proper centriole number.

Centrioles are small barrel-shaped organelles that play critical roles in organizing centrosomes and cilia[1]. Although the structure of centrioles can vary among species and cell types, they all share a few common architectural features. Centrioles exhibit a nine-fold rotational symmetry marked by the presence of microtubule singlets, doublets or triplets that form the outer wall. The centriole is fortified internally by protein scaffolds that direct the overall organization and impart structural integrity to the centriole[2,3]. These include the ubiquitous cartwheel found in the centrioles of worms, vertebrates and many other species[4–7] and the inner scaffold found in the central and distal regions of vertebrate and protist centrioles[3].

Centrioles are typically found in pairs with the younger or daughter centriole positioned perpendicular to the long axis of the older or mother centriole. In dividing cells centrioles recruit a surrounding matrix of pericentriolar material (PCM) to form centrosomes, the cell's microtubule-organizing centers[8]. In such cells centrioles are under strict numerical control so that one pair (and thus one centrosome) is present at the beginning of the cell cycle and two pairs (and thus two centrosomes) are present as the cell enters M phase when each centrosome will establish one pole of the bipolar mitotic spindle[9]. A key determinant of numerical control is a single round of duplication that takes place during S phase where one and only one daughter is assembled next to each mother centriole. Defects in numerical control can lead to centriole amplification, which is a common feature of cancer cells and has been shown to induce tumorigenesis in mice[10,11]. Cancer-related centriole amplification can arise through several pathways including overduplication[10] or through structural defects that lead to the fragmentation of existing centrioles[12].

[1]Laboratory of Biochemistry and Genetics, National Institutes of Diabetes and Digestive and Kidney Diseases, National Institutes of Health, Bethesda, MD, USA. [2]Laboratory of Structural Cell Biology, National Heart, Lung, and Blood Institute, National Institutes of Health, Bethesda, MD, USA. [3]Department of Molecular and Cellular Biology, University of Geneva, Geneva, Switzerland. [4]National Institute of Arthritis and Musculoskeletal and Skin Diseases, National Institutes of Health, Bethesda, MD, USA. [5]These authors contributed equally: Jason A. Pfister, Lorenzo Agostini. ✉e-mail: naoko.mizuno@nih.gov; kevino@nih.gov

*Caenorhabditis elegans* has long been used as a model to study centriole assembly[8]. As in other species, *C. elegans* utilizes a conserved set of core centriole assembly proteins. The master regulator of duplication is a polo-like kinase named ZYG-1[13], a relative of vertebrate Plk4. ZYG-1 functions to recruit the cartwheel components SAS-5 (STIL) and SAS-6[14–17] through a direct physical interaction[18]. Phosphorylation of SAS-5 on serine 10 drives the recruitment of SAS-4, a relative of vertebrate CPAP[19,20], which assembles the microtubule wall, locking the other components into place[21,22].

Compared to most other cellular constituents, centrioles exhibit unusual stability. After incorporation into mammalian centrioles, tubulin remains stably associated even under conditions that depolymerize cytoplasmic microtubules[23]. In *C. elegans* embryos, SAS-4 and SAS-6 do not exhibit significant—if any—loss from centrioles over the course of seven or more cell cycles[24]. Centriole stability however is developmentally regulated. In certain tissues centrioles are actively eliminated. These tissues include the female germ line of many species, as well as polyploid cells and syncycia[25,26]. How centrioles achieve their stability and how they disassemble in response to developmental cues are still not well understood. One protein that plays a key role in centriole stability and elimination in *C. elegans* is the C2 domain-containing protein SAS-1[27]. SAS-1 is a homolog of vertebrate C2CD3 which is required for normal centriole morphology[28–30]. SAS-1 localizes to the central tube, a protein scaffold inside the microtubule outer wall[7]. Not only is SAS-1 required for the stability of embryonic centrioles[27], its loss from centrioles in the female germ line is an initiating event in centriole elimination[25]. Furthermore, *sas-1* mutants exhibit premature elimination of centrioles.

A complete understanding of how centriole stability is established, maintained, and regulated will require the identification and characterization of all factors involved. The Sjögren's Syndrome Nuclear Antigen 1 (SSNA1) protein is a small filamentous microtubule-binding protein that localizes to centrosomes, basal bodies and sites of axon branching[31–34]. In vitro, SSNA1 has been shown to induce branching of microtubules during polymerization[31], as well as stabilize dynamic microtubules and detect defects in the microtubule lattice[35]. While SSNA-1 is known to be required for proper cell division in human cells and Chlamydomonas[32,34], its precise role in this process has yet to be defined.

In this study we identify the *C. elegans* ortholog of SSNA1 as a constituent of centrioles and centriole satellite-like structures. SSNA-1 localizes in close proximity to the central tube where SAS-1 resides and genetically interacts with SAS-1. A complete knockout of SSNA-1 results in embryonic lethality marked by the appearance of multipolar spindles. We provide direct evidence that the extra spindle poles arise via fragmentation of existing centrioles, revealing that SSNA-1, like SAS-1, is required for the structural integrity of centrioles. In conjunction with a parallel in vitro study[36], we show that SSNA-1 likely executes this function via its ability to self-assemble into filaments that bind microtubules.

## Results

### *C. elegans* possesses an SSNA1 ortholog
Using the Alliance of Genome Resources website (www.alliancegenome.org), we identified a putative *C. elegans* ortholog of *Xenopus tropicalis* SSNA1 (Fig. 1A), encoded by the open reading frame T07A9.13. Consistent with this finding, an earlier study also identified T07A9.13 as a putative ortholog of SSNA-1[37]. T07A9.13 is an uncharacterized gene found at the end of the three gene operon CEOP4622 and encodes a 12.5 kDa protein of 105 amino acid residues. T07A9.13 is preceded by two genes (*rps-24* and T07A9.14) that are predicted to be structural constituents of the ribosome and orthologs of the human RPS24 protein. Sequence based alignment revealed that the protein product of T07A9.13 shows ~25% identity and ~45% similarity to other SSNA1 proteins (Fig. 1A, B).

Previous studies have established that SSNA1 orthologs are small (~14 kDa) proteins that are almost entirely alpha-helical[31,34,38–40]. Consistent with T07A9.13 encoding the *C. elegans* SSNA1 homolog, an AlphaFold structural prediction (Q95X71_CAEEL) and our parallel study[36] indicates that the protein encoded by T07A9.13 also forms a single extended alpha helix (Fig. 1C). SSNA1 orthologs have been shown to form bundled filaments that bind microtubules and induce microtubule branching[31,39–41]. Similarly, the T07A9.13 protein behaves in an identical manner in vitro[36]. Based on its homology, structural similarity, and shared biochemical activity, we conclude that T07A9.13 is the *C. elegans* ortholog of SSNA1. In accordance with *C. elegans* genetic nomenclature, we will refer to the gene and its protein as *ssna-1* and SSNA-1, respectively.

### Loss of SSNA-1 results in maternal-effect embryonic lethality
Thus far, the consequences of SSNA1 loss have been explored in single-celled organisms or in tissue culture cells. Knockout of SSNA1 in either Trypanosomes[39] or Toxoplasma[38] does not produce an obvious phenotype, suggesting that SSNA1 is not essential in protozoans. Conversely, RNA interference or antisense-mediated knockdown of SSNA1 in tissue culture cells[32] or *Chlamydomonas*[34], respectively, results in a multinucleate phenotype. While these later studies suggest that SSNA-1 plays a role in cell division, neither study explored a deeper understanding of the defects observed.

To precisely define the biological roles played by SSNA-1, we investigated the effects of a complete knockout of SSNA-1 in a multicellular organism by using CRISPR-Cas9 to precisely remove the SSNA-1 ORF in *C. elegans*. We refer to the knockout allele as *ssna-1(bs182)* or more simply *ssna-1(Δ)*. We found that animals homozygous for *ssna-1(Δ)* were viable and fertile. While most mutant animals appeared morphologically wild-type, a small fraction born from homozygous mothers exhibited vulval defects or were somewhat shorter than wild-type[36]. Furthermore, the mutant animals did not exhibit an obvious defect in motility, indicating essentially normal neuromuscular function. Animals homozygous for *ssna-1(Δ)* did however exhibit a highly penetrant phenotype: a precipitous drop in the viability of their offspring. We found that only about 30% of the progeny of *ssna-1(Δ)* animals survived embryogenesis at 20 °C, and just 7.5% survived at the elevated temperature of 25 °C (Fig. 1D, E). We independently confirmed this phenotype in a strain engineered to contain a premature stop codon in the *ssna-1* coding sequence (*ssna-1(bs222[G7X])*); such mutant animals exhibited a comparable reduction in embryonic viability (Fig. 1D). As phenotypes were not observed in *ssna-1(Δ)/+* heterozygotes, the deletion behaves in a recessive manner. Finally, to show that the embryonic lethal phenotype specifically arises from a loss of *ssna-1*, we crossed a wild-type *ssna-1* transgene into the *ssna-1(Δ)* strain and found that it completely rescued the embryonic lethality (Fig. 1F).

We next investigated if filament formation was required for SSNA-1 function. We found that small deletions of the N- or C-terminus (Fig. 1D, alleles Δ2-17, Δ2-18, Δ2-22, and Δ100-105) also resulted in an embryonic lethal phenotype that was at least as severe as *ssna-1(Δ)*. As these N- and C-termini residues are required for self-assembly of SSNA-1 filaments[31,36], these results suggest that the ability to form oligomers is required for the in vivo function of SSNA-1.

Embryonic lethality can arise due to defects in the maternal and/or paternal contributions to the embryo. We thus tested if the embryonic lethal phenotype of the *ssna-1(Δ)* mutant was due to a maternal and/or paternal loss of function. When *ssna-1(Δ)* males were mated to wild-type *fem-1(hc17)* females, no reduction in embryonic viability was observed, indicating the absence of a paternal defect (Fig. 1G). In contrast, mating wild-type males to *ssna-1(Δ), fem-1(hc17)* females resulted in a significant reduction in embryonic viability (Fig. 1G) indicating a strong maternal effect. As the level of viability among the offspring of *ssna-1(Δ)* mothers mated to either wild-type or

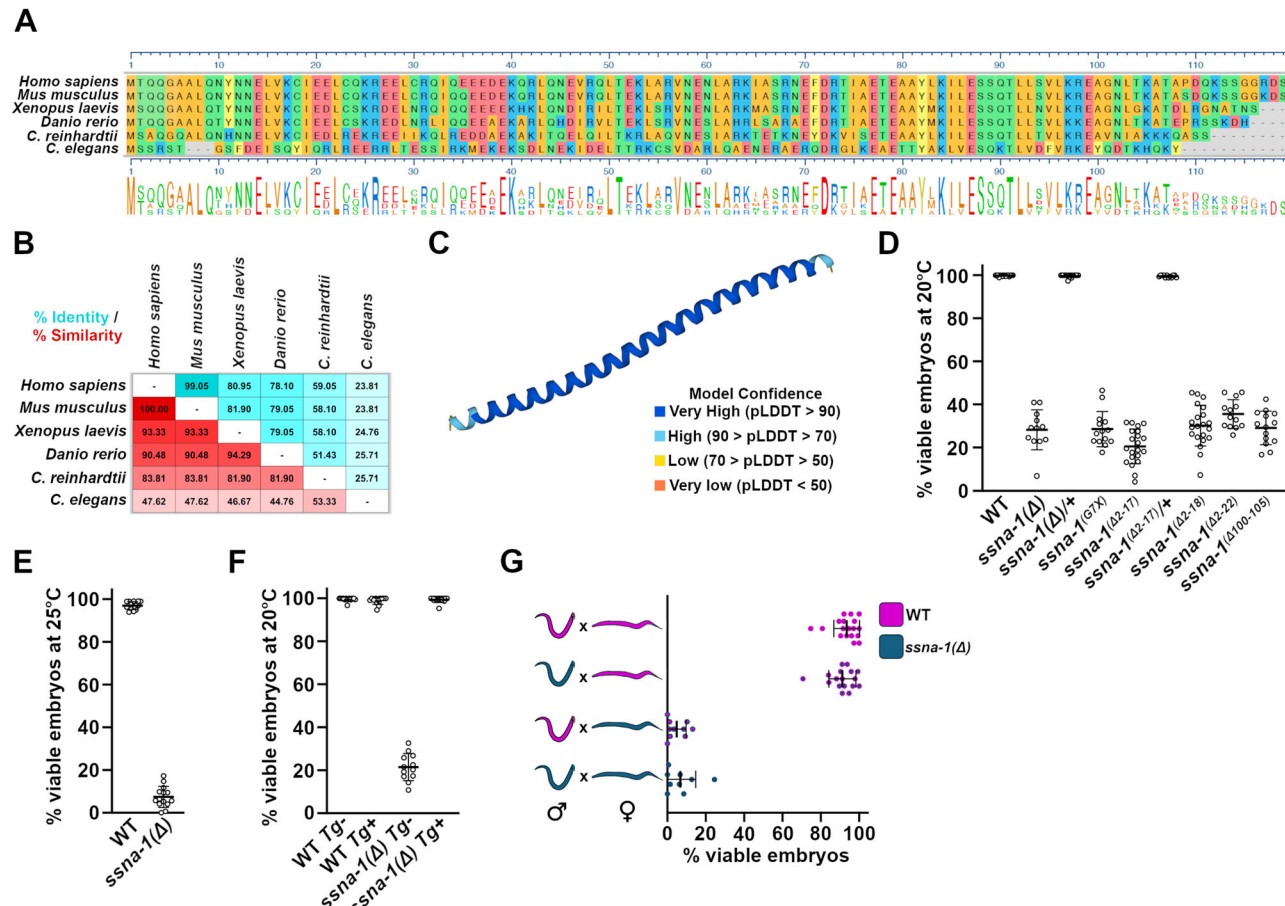

**Fig. 1 | *C. elegans* possesses an SSNA-1 ortholog that plays a critical role in embryogenesis. A** Sequence alignment of various SSNA-1 orthologs. The amino acids are colored as follows: blue and red for negatively and positively charged residues, respectively, green for uncharged residues, and gold for hydrophobic residues. **B** Table showing percent identity (blue) and percent similarity (red) of SSNA-1 orthologs. **C** AlphaFold structural prediction of T07A9.13. **D** Embryonic viability of wild-type and *ssna-1* mutant strains at 20 °C. Each datapoint represents progeny of a single hermaphrodite. Mean and SD are shown. *n* = 21 (WT), 12 (*ssna-1(Δ)*), 14 (*ssna-1(Δ)/+*), 14 (*ssna-1^(G7X)*), 21 (*ssna-1^(Δ2-17)*), 12 (*ssna-1^(Δ2-17)/+*), 21 (*ssna-1^(Δ2-18)*), 13 (*ssna-1^(Δ2-22)*), 14 (*ssna-1^(Δ100-105)*). **E** Embryonic viability of wild-type and *ssna-1(Δ)* strains at 25 °C. Each datapoint represents progeny of a single hermaphrodite. Mean and SD are shown. *n* = 14(WT), 14(*ssna-1(Δ)*). **F** Embryonic viability of wild-type and *ssna-1(Δ)* strains with or without the *ssna-1* transgene (Tg). Note that when expressed in the wild type, the transgene does not affect viability, but when expressed in the *ssna-1(Δ)* mutant it fully rescues the embryonic lethal phenotype. Each datapoint represents progeny of a single hermaphrodite. Mean and SD are shown. *n* = 13 (WT Tg−), 13 (WT Tg+), 12 (*ssna-1(Δ)* Tg−), 13 (*ssna-1(Δ)* Tg+). **G** SSNA-1 is maternally required. Embryonic viability among the progeny of wild-type (magenta) or *ssna-1(Δ)* (blue) males and wild-type or *ssna-1(Δ), fem-1(hc17)* females. Note that embryonic lethality correlates with the genotype of the mother and not the father. No significant differences were observed in viability among the progeny sired by wild-type or mutant males. Each datapoint represents progeny of a single female. Mean and SD are shown. *n* = 18 (WT × WT), 17 (*ssna-1(Δ)* males × WT), 10 (WT × *ssna-1(Δ)* females), 9 (*ssna-1(Δ)* × *ssna-1(Δ)*). Source data are provided as a Source Data file.

*ssna-1(Δ)* males was the same (Fig. 1G), the embryonic lethal phenotype results strictly from the loss of *ssna-1* in the maternal germline.

## SSNA-1 localizes to centrioles and satellite-like structures
Orthologs of SSNA-1 have previously been localized to centrosomes, basal bodies, and dynamic microtubules[31,32,34,35,38,39]. To localize SSNA-1 by immunofluorescent staining, we used CRISPR-Cas9 genome engineering to modify the endogenous gene so that the protein contained a small SPOT epitope tag at the C-terminus. The strain expressing SSNA-1::SPOT lacks an embryonic lethal phenotype indicating that the tagged protein is functional (Fig. S1A). Upon analyzing SSNA-1::SPOT, we found that it co-localized with the centriole marker SAS-4 throughout embryogenesis, indicating that the protein is associated with centrioles (Fig. 2A, B). Additionally, we modified the endogenous *ssna-1* gene with the four amino acid C-terminal C-tag, and by immunofluorescence staining found that it exhibited an identical localization pattern[36]. From fertilization through the end of the first division, SSNA-1 was solely localized with centrioles and was not enriched anywhere else, including nuclei (Fig. 2A). However, near the end of the first mitosis, a few additional foci began to appear in the vicinity of the centrioles (Fig. 2A, arrowheads in Telophase panel). During the second cell cycle SSNA-1::SPOT remained associated with centrioles (Fig. 2B), but after nuclear envelope breakdown, there was a dramatic increase in the number and size of foci surrounding the centrosomes (Fig. 2B Metaphase-Telophase). This localization pattern was also seen using an antibody raised to the full-length endogenous protein (Fig. S1B). We refer to these newly identified centrosome-associated bodies as satellite-like structures as they are reminiscent of centriole satellites observed in vertebrate cells and *Drosophila*[42,43].

We next co-localized SSNA-1::SPOT with the PCM marker SPD-5 that is tagged at its endogenous locus with GFP. We found that the satellite-like structures reside outside of the PCM and therefore outside of the centrosome (Fig. 2C). At the end of mitosis, the PCM is disassembled in part through mechanical forces that fragment the PCM into "packets"[44]. At this time the SSNA-1::SPOT puncta also spread out and dispersed, assuming a distribution similar to GFP::SPD-5

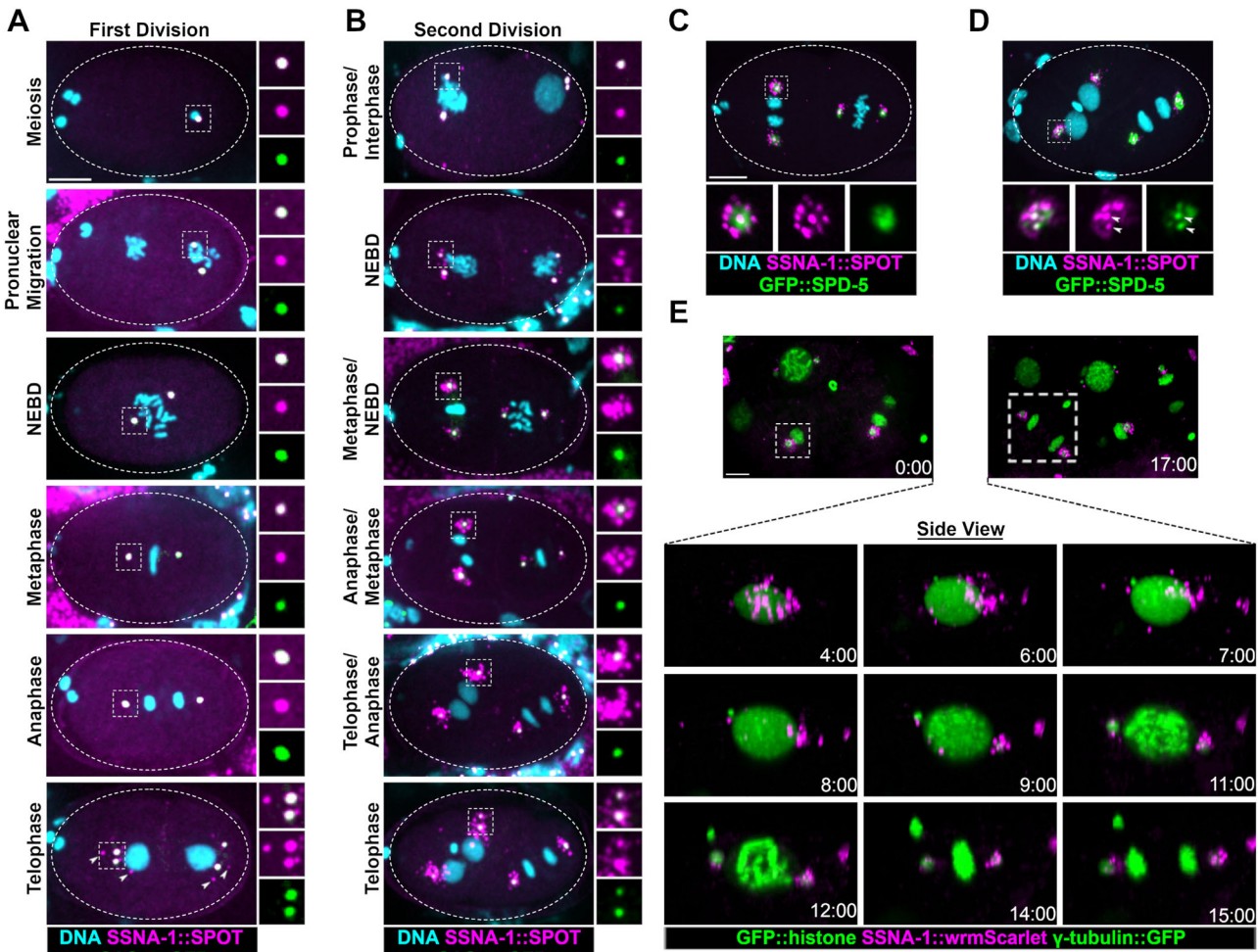

**Fig. 2 | SSNA-1 is a component of the centriole and centriole satellite-like structures. A** Immunofluorescence images of SSNA-1::SPOT and SAS-4::GFP during the first embryonic division. SSNA-1 is found only at centrioles from meiosis through anaphase of the first cell cycle. During first telophase, additional weaker SSNA-1::SPOT foci can be seen (arrowheads). Scale bar = 10 μm and applies to (**A** and **B**). Enlargements are 2× magnifications. Experiment was repeated three times with identical results. **B** During the second cell cycle SSNA-1::SPOT remains centriolar but beginning around nuclear envelope breakdown (NEBD), SSNA-1::SPOT assembles into satellite-like structures that surround the centriole. Enlargements are 2× magnifications. Experiment was repeated three times with identical results. **C, D** Immunofluorescence image showing co-localization of SSNA-1::SPOT relative to the PCM marker GFP::SPD-5. Twenty-six two-cell embryos were

examined with identical results. **C** During anaphase, SSNA-1::SPOT satellite-like structures localize outside of the PCM marked by GFP::SPD-5. Scale bar = 10 μm and applies to (**C** and **D**). **D** During telophase when the PCM breaks up into packets, SSNA-1::SPOT satellite-like structures remain distinct from SPD-5 packets. However, a portion of both SSNA-1::SPOT and GFP::SPD-5 co-localize at the centriole (arrowheads in enlargements). **E** Frames from a time-lapse recording of an embryo expressing SSNA-1::wrmScarlet, γ-tubulin::GFP, and GFP::Histone H2B. First and last frame are at top. SSNA-1::wrmScarlet reorganizes during division of the ABpl cell (boxed). Time-lapse images reveals that the satellite-like structures disperse over the nucleus during interphase and accumulate at centrosomes as cells enter mitosis. Time is shown as mm:ss. Scale bar = 10 μm. A total of 40 embryos were imaged with similar results.

(Fig. 2D). Despite similar distributions, the SSNA-1 satellite-like structures did not co-localize with the SPD-5 packets (Fig. 2D). However, centriole-associated SSNA-1 did co-localize with the pool of SPD-5 remaining in close association with centrioles (Fig. 2D arrowheads).

SSNA-1::SPOT was also detected at the centrioles of sperm cells and immature female germ cells, although satellite-like structures were not detected in these cells (Fig. S1C). Centrioles are eliminated during oogenesis, in a process that begins in early diplotene[25,45]. As a result, centrioles are absent in mature oocytes. Accordingly, we did not detect SSNA-1::SPOT foci in mature oocytes and in fact found that SSNA-1 signal is lost by mid-to-late pachytene, prior to the core centriolar protein SAS-4 (Fig. S1C, columns 3 and 4). This suggests that removal of SSNA-1 might be an early event in the elimination program. SSNA-1::SPOT staining was also evident at the base of amphid cilia in the head of young (L1/L2 stage) larvae (Fig. S1D) where it localized just distally to the acentriolar MTOC marked by SPD-5[46,47]. As centrioles have been reported to be absent in these cells[48,49], the SSNA-1::SPOT foci might

represent either centriole remnants or a non-centriole-related structure.

To determine if SSNA-1 is stably associated with centrioles, we mated SSNA-1::SPOT expressing males to hermaphrodites lacking the SPOT epitope tag. We then stained the embryonic progeny with the anti-SPOT nanobody. If SSNA-1::SPOT is a stable component of centrioles we would expect to be able to detect the two sperm derived centrioles, but if SSNA-1::SPOT exchanges with the cytoplasmic pool of unlabeled SSNA-1, the centriole-associated SPOT signal should fade over time, eventually being lost. Indeed, we clearly detected two SPOT-tagged centrioles in multicellular embryos (Fig. S1E), revealing that SSNA-1 is stably associated with centrioles over the course of many cell cycles.

To analyze the dynamic behavior of the satellite-like structures, we used CRISPR-Cas9 to tag the C-terminus of SSNA-1 with wrmScarlet (*ssna-1(bs243)*). Although this resulted in a ~ 30% decrease in embryonic viability (Fig. S1A) indicating that the tag interfered with the

function of SSNA-1, SSNA-1::wrmScarlet still localized to centrioles and satellite-like structures (Fig. 2E). Time-lapse recording showed that the satellite-like structures dispersed around nuclei in interphase but progressively accumulated around centrosomes as cells progressed into mitosis. (Movie S1).

## SSNA-1 localizes inside and in close proximity to the micro-tubule wall of centrioles

To map the position of SSNA-1 within the centriole at high resolution, we performed Ultrastructure Expansion Microscopy (U-ExM) from gonadal spreads[7,50,51] using a series of native or endogenously tagged reference proteins. Mapping SSNA-1 relative to tubulin, which defines the centriole's outer wall, and SAS-4, we found that HA::SAS-4 staining coincides with the centriole's outer wall with a mean diameter of approximately 100 nm (Fig. 3A, D, E). This close association between SAS-4 and the outer wall is consistent with previously published work[7]. In contrast, SSNA-1 appeared as a ring in cross section with a mean diameter of 72 nm that lies just inside the outer wall (Fig. 3A, D, E). In transverse orientations, SSNA-1 could be seen to run along the entire length of the centriole (Fig. 3A). We also mapped SSNA-1 relative to SAS-6, a component of the cartwheel[7]. SAS-6::HA was found as a dot in the center of the centriole barrel with a mean diameter of 61 nm. SAS-6::HA was found within the SSNA-1 ring (Fig. 3B, D, E) and thus SSNA-1 resides between the C-terminus of SAS-6 and the outer centriole wall. In the course of our analysis, we also observed emergent daughter centrioles decorated with HA::SAS-4 and SAS-6::HA (arrowheads in Fig. 3A, B). Interestingly, neither daughter centriole stained for SSNA-1, indicating that SSNA-1 is recruited to nascent centrioles after SAS-4 and SAS-6. We then determined the relative localization of ZYG-1 and SSNA-1 and found that ZYG-1 was more peripheral localizing as a set of foci or as a partial ring just outside the outer wall (Fig. 3C–E).

Our U-ExM results indicate that SSNA-1 localizes near the central tube where the centriole-stabilizing factor SAS-1 resides[7]. We thus mapped the position of SSNA-1 relative to SAS-1::3 × Flag and found that they closely overlap (Fig. 3F). We also found that this arrangement was maintained in embryonic centrioles indicating that the relative position of these two proteins does not change during development (Fig. 3G). In summary, we conclude that SSNA-1 is an integral component of the centriole that localizes in close association with SAS-1.

## Microtubule binding likely contributes to SSNA-1 localization and function

We next sought to determine if SSNA-1's localization to centrioles and centriole satellite-like structures is dependent on its oligomerization and/or microtubule binding activity. We began by examining embryos expressing a SPOT-tagged version of SSNA-1 lacking amino acids 2–18 (*ssna-1(bs137)*). As shown in Fig. 1D, deletion of amino acids 2–18 results in a complete loss of function, and in vitro, a protein that lacks these same residues fails to form filaments and has a severely reduced ability to bind microtubules[36]. By immunofluorescent staining, SSNA-1$^{\Delta2-18}$::SPOT failed to localize to either centrioles or centriole satellite-like structures (Fig. S2A), suggesting that filament formation and/or microtubule binding is essential for its localization. It should be noted however that we can't detect SSNA-1 by immunoblotting, and thus, do not know if SSNA-1$^{\Delta2-18}$::SPOT is expressed. We next examined the localization of SSNA-1$^{R18E}$::SPOT. In vitro analysis showed that the residue R18 is important for stable microtubule binding, but its mutation to glutamic acid does not affect filament formation[36]. Embryos expressing SSNA-1$^{R18E}$ exhibit a temperature-sensitive phenotype whereby ~90% of the embryos are viable at 20 °C while only ~65% are viable at 25 °C[36]. Interestingly, at 20 °C we found that SSNA-1 was completely absent from the satellite-like structures but could still be detected at centrioles (Fig. S2B). Using quantitative immuno-fluorescence, we found that recruitment of SSNA-1 to centrioles was

also affected by the R18E mutation; compared to wild-type SSNA-1, the level of SSNA-1$^{R18E}$ at centrioles was markedly reduced (Fig. S2C). We also noticed that SSNA-1$^{R18E}$::SPOT often failed to associate with sperm centrioles before and immediately after fertilization (Fig. S2D and E). Thus, our data suggests that microtubule binding plays a role in recruiting SSNA-1 to both centrioles and centriole satellite-like structures, and that microtubule binding is also likely required for SSNA-1 function. Finally, this data also suggests that the pool of SSNA-1 localized to the centriole satellite-like structures does not play a critical role during embryogenesis.

## Deletion of SSNA-1 results in multipolar spindle formation

We next set out to determine the underlying cause of the embryonic lethal phenotype of *ssna-1(Δ)* mutants. We performed live imaging of control and *ssna-1(Δ)* embryos expressing GFP::histone H2B, mCherry::β-tubulin, and GFP::SPD-2 at 25 °C. Wild-type embryos, as expected, assembled only bipolar spindles during each mitotic division (Fig. 4A–C). Similarly, through the first two cell cycles, nearly all of the *ssna-1(Δ)* embryos appeared normal and assembled bipolar spindles. However, 89.7% of *ssna-1(Δ)* embryos assembled at least one multipolar spindle between the second and fourth cell cycles (Fig. 4A, B), with the great majority of multipolar spindles appearing after the two-cell stage (Fig. 4C). Most of the multipolar spindles had three poles (89.6%) with the remainder having four (Fig. 4D). Interestingly, we also found that when grown at 25 °C embryos expressing SSNA-1$^{R18E}$::SPOT also exhibit a multipolar spindle phenotype, suggesting that microtubule-binding is important for SSNA-1's centrosome function (Fig. S2F). Embryos lacking SSNA-1 also exhibit a less frequently observed defect: detachment of a centrosome from the nuclear envelope. This occurs in ~15% of embryos (Fig. 4B and Fig. S3A). When the multipolar spindle phenotype was examined on a cell-by-cell basis, it became obvious that almost no such defects were present before the 4-cell stage (Fig. 4E and Fig. S3B). Also, the multipolar spindle phenotype was not biased toward a particular lineage or cell type, as all cells of four- and eight-cell stages were similarly affected (Fig. S3B). Thus, spindle defects appear to arise stochastically beginning after the second division.

To determine if the extra spindle poles contain centrioles we examined *ssna-1(Δ)* embryos expressing endogenously tagged ZYG-1::SPOT and co-stained with an antibody recognizing endogenous SAS-4 (Fig. 4F). All poles of multipolar spindles stained positive for both centriole components indicating that the additional poles contain bona fide centrioles. Additionally, we found that the extra poles also contained the centriole components SAS-1, SAS-5, SAS-6, and SAS-7 (Fig. S3C–E). These results suggest that the multipolar spindle phenotype arises as a consequence of some type of centriole amplification defect, such as overduplication, reduplication, or centriole fragmentation.

## ZYG-1 and SSNA-1 genetically and physically interact

We first sought to determine if overduplication (i.e., the formation of extra daughter centrioles) might be occurring in the *ssna-1(Δ)* mutant. It has been shown that increased expression of core centriolar proteins, such as ZYG-1/Plk4, SAS-5/STIL or SAS-6, can drive overduplication[52–56]. To determine if such increases in centriole protein levels might be responsible for the centriole amplification defect of *ssna-1(Δ)* mutants we examined the levels of these proteins by quantitative immunoblotting and found that the total levels of SAS-5 and SAS-6 are not elevated in the *ssna-1(Δ)* mutant (Fig. S4A). We then used quantitative immunofluorescence to examine the *ssna-1(Δ)* mutant for increased levels of ZYG-1, SAS-5, and SAS-6 at centrioles. We focused on the second round of centriole assembly that occurs near the end of the first mitotic division, as a defect in this round of centriole assembly would produce the additional poles that appear at the four-cell stage. Given the striking centriole amplification defect of

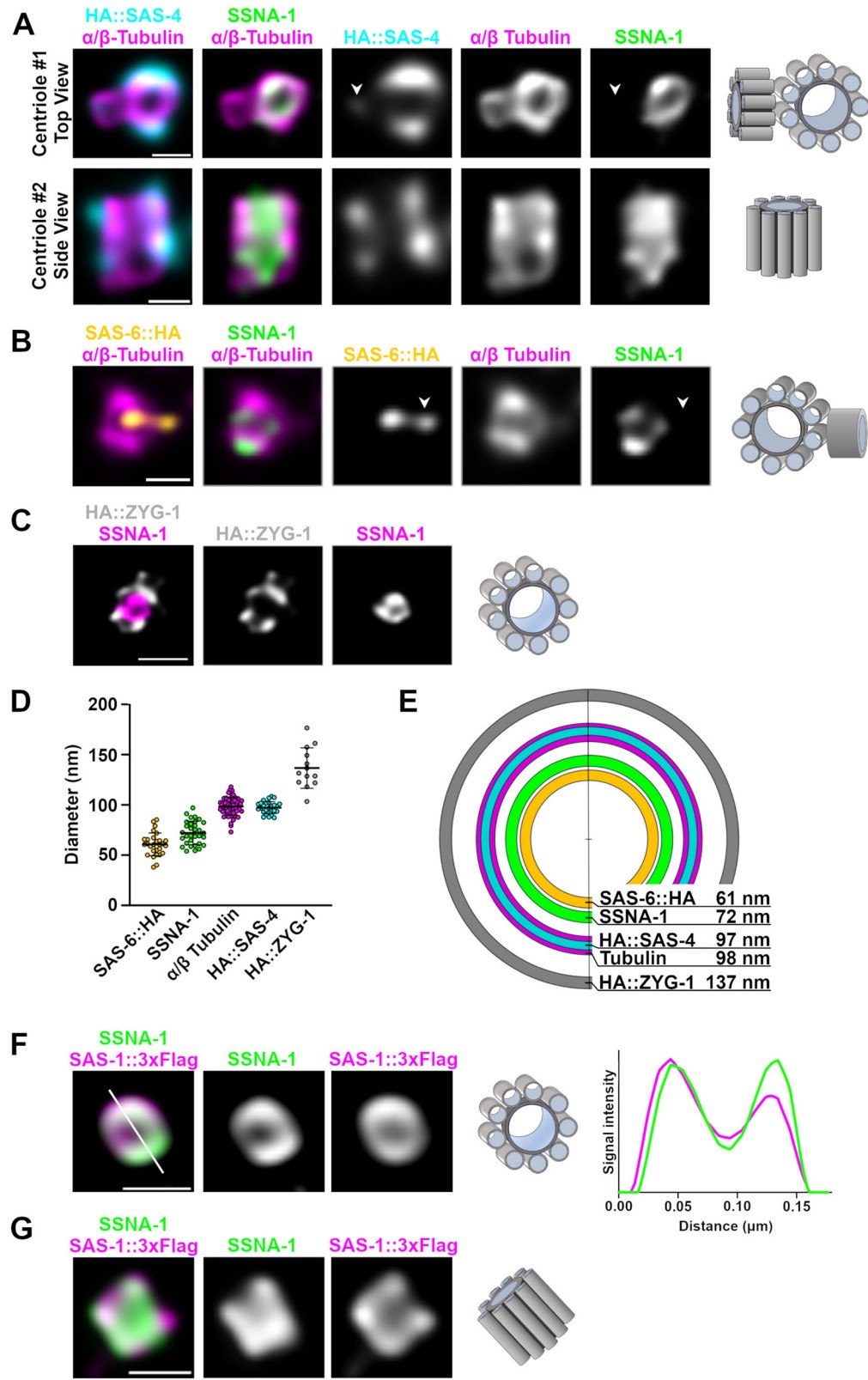

ssna-1(Δ) mutants, we were surprised to find that the centriolar levels of ZYG-1, SAS-5, and SAS-6 were unaltered in the mutant (Fig. S4B). We also analyzed the centrosome-associated levels of SPD-2 and SPD-5 and found no difference between wild-type and ssna-1(Δ) mutant embryos (Fig. S4C). In summary, we did not observe any increases in the levels of centriole assembly factors that could explain an overduplication defect.

Although the levels of centriole proteins were unaltered, it remained possible that centrioles were overduplicating in the ssna-1(Δ) mutant. Thus, we turned to a genetic approach to test for over-duplication in the ssna-1(Δ) mutant. We reasoned that if the extra centrioles in ssna-1(Δ) mutants arise due to centriole overduplication, we should be able to rescue this defect by partially inhibiting the centriole assembly pathway. For this purpose, we utilized the

**Fig. 3 | Centriolar SSNA-1 localizes between the microtubule-containing outer wall and the cartwheel. A–C** Ultrastructure Expansion Microscopy (U-ExM) of centrioles from gonad spreads. The schematic shown to the right of each set of images depicts the orientation and configuration of centrioles. **A** Co-localization of SSNA-1, HA::SAS-4, and α/β-tubulin from two distinct centrioles showing a top view (centriole #1) and side view (centriole #2). SSNA-1 is localized inside the microtubule outer wall. Arrowhead indicates emergence of a daughter centriole that stains weakly for HA::SAS-4 but not SSNA-1. Scale bars = 100 nm. **B** Co-localization of SSNA-1, SAS-6::HA, and α/β-tubulin. SSNA-1 is localized outside of the cartwheel defined by SAS-6::HA. Arrowhead indicates emergence of a daughter centriole that stains positive for SAS-6::HA but not SSNA-1. Scale bar = 100 nm. **C** Co-localization

of SSNA-1 and HA::ZYG-1. Scale bar = 200 nm. **D** The measured diameters of each protein are plotted with each point representing a measurement from a single centriole. The mean and standard deviation are shown. *n* = 28 (SAS-6::HA), 32 (SSNA-1), 55 (α/β-Tubulin), 27 (HA::SAS-4), and 13 (HA::ZYG-1). **E** Diagram indicating the position of each protein within the centriole. SSNA-1 localizes between SAS-6::HA and α/β-tubulin. **F** Co-localization of SSNA-1 and SAS-1::3 × Flag in centrioles from gonad spreads. Scale bar = 100 nm. To the right is a signal intensity profile plot of SSNA-1 and SAS-1::3 × Flag. **G** Co-localization of SSNA-1 and SAS-1::3 × Flag in embryonic centrioles. Scare bar = 100 nm. Source data are provided as a Source Data file.

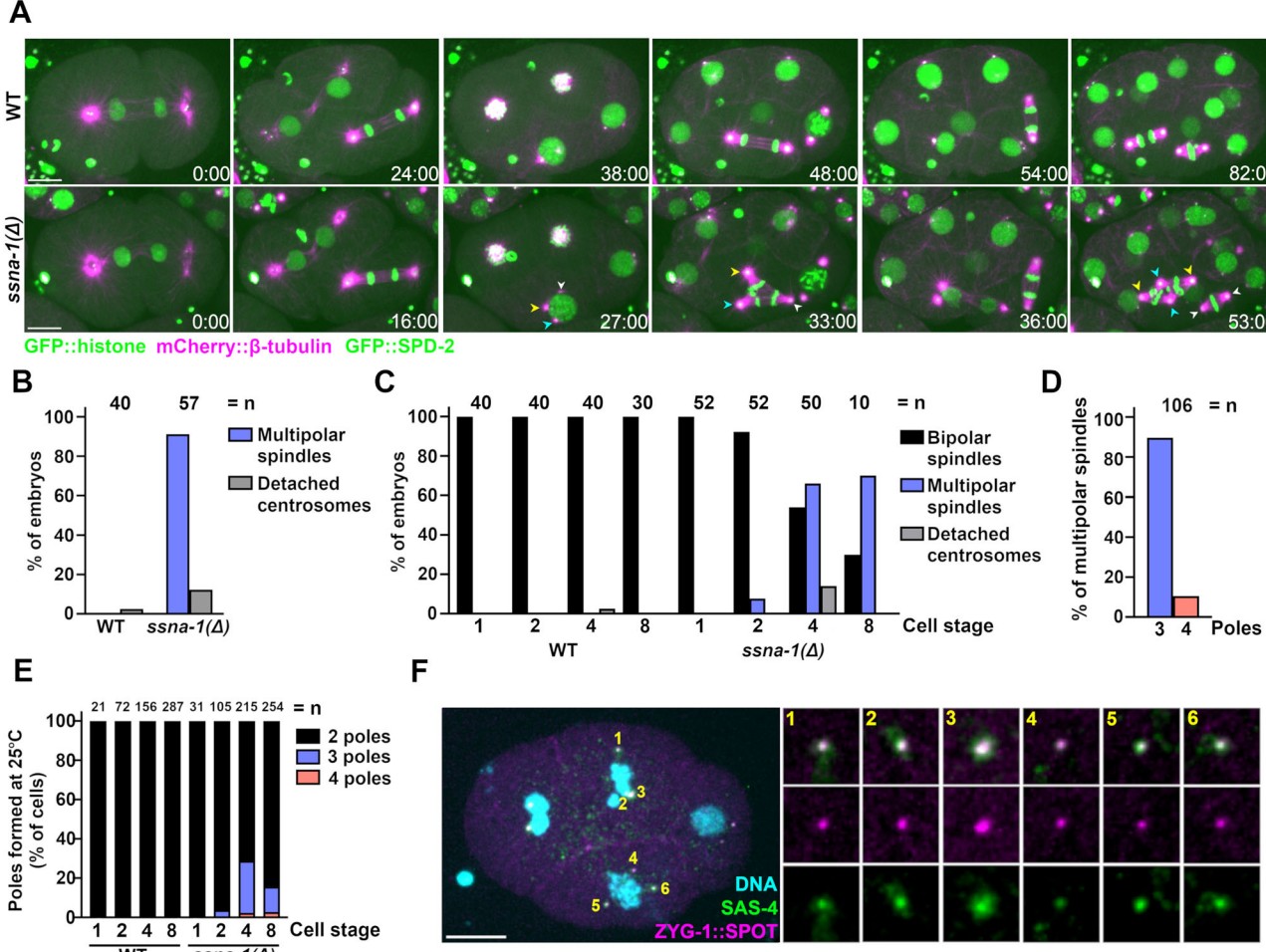

**Fig. 4 | Deletion of SSNA-1 results in excess centrioles. A** Time-lapse images of the first four rounds of division of either wild-type or *ssna-1(Δ)* embryos expressing GFP::histone, mCherry::β-tubulin, and GFP::SPD-2. While wild-type embryos form only bipolar spindles, *ssna-1(Δ)* embryos form multipolar spindles beginning around the four-cell stage. During the third round of division in the *ssna-1(Δ)* embryo (*t* = 27:00), multiple centrosomes highlighted by cyan, yellow, and white arrowheads appear. These form a multipolar spindle in the EMS cell (*t* = 33:00). Each centrosome is capable of duplication as shown in the subsequent daughter cells E (white arrowheads) and MS (cyan and yellow arrowheads). Scale bar = 10. **B** Percent of embryos displaying a multipolar spindle or detached centrosome defect. (*n* = number of embryos scored). **C** Percent of embryos at each cell stage

displaying at least one multipolar spindle or a detached centrosome defect. The percent of embryos displaying no defect (bipolar spindles only) are represented by black bars. (*n* = number of embryos scored). **D** Percentage of tripolar and tetrapolar spindles among all multipolar spindles (*n* = spindles scored). **E** Percentage of cells with a bi-, tri- or tetrapolar spindle at each embryonic cell stage (*n* = number of cells scored). **F** An immunofluorescent image of a four-cell stage *ssna-1(Δ)* embryo with a multipolar spindle in both the ABp (1–3) and EMS (4–6) cells. Note that all spindle poles are positive for both ZYG-1::SPOT and SAS-4 indicating the extra spindle poles contain centrioles. Scale bar = 10 μm. Nineteen embryos with multipolar spindles were examined with all poles of all multipolar spindles staining positive for each marker. Source data are provided as a Source Data file.

temperature-sensitive *zyg-1(it25)* mutation whose activity can be progressively inhibited by incrementally increasing temperature[57]. Using a semi-permissible temperature of 22.5 °C, we determined the levels of embryonic viability among three strains: *zyg-1(it25), ssna-1(Δ), and zyg-1(it25); ssna-1(Δ)* mutants. At this temperature, *zyg-1(it25)* embryos showed a slight decrease in viability (87.8%), indicating a partial

inhibition of centriole assembly, while *ssna-1(Δ)* embryos exhibited 22% viability (Fig. 5A). Contrary to our expectations, the *zyg-1* mutant did not suppress the embryonic lethality of the *ssna-1* deletion. In fact, the *zyg-1(it25); ssna-1(Δ)* double mutant produced no viable progeny at all (Fig. 5A). The observed level of lethality was not the result of the combined effects of both single mutations as we would have expected

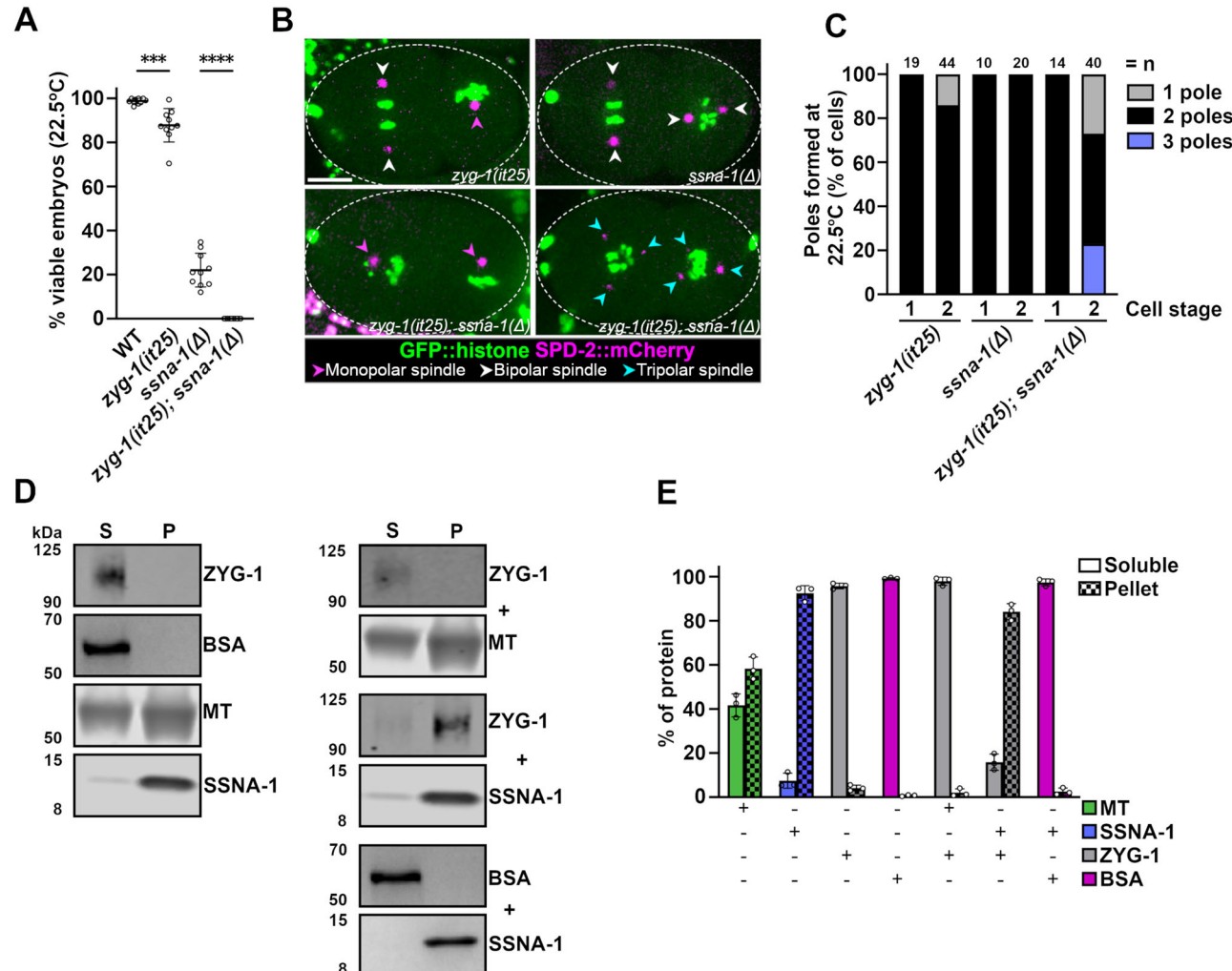

**Fig. 5 | SSNA-1 functions during centriole assembly. A** Embryonic viability of progeny from wild type, *zyg-1(it25), ssna-1(Δ)*, and *zyg-1(i25); ssna-1(Δ)* strains at 22.5 °C. Each data point represents the progeny of a single hermaphrodite. Mean and SD are shown. *n* = 10 (WT), 10 (*zyg-1(it25)*), 10 (*ssna-1(Δ)*), 10 (*zyg-1(it25); ssna-1(Δ)*). ****p* = 0.0003, *****p* < 0.0001 as determined with a one-way ANOVA with Tukey's multiple comparisons test. **B** Single frames taken from time-lapse recordings of the indicated strains expressing GFP::histone and SPD-2::mCherry. Each image shows a two-cell stage embryo. A few *zyg-1(it25)* embryos exhibit monopolar spindles (magenta arrowhead), while *ssna-1(Δ)* embryos only possess bipolar spindles (white arrowheads). The *zyg-1(it25); ssna-1(Δ)* double mutant embryos exhibit a mixed phenotype, with monopolar spindles (magenta arrowheads) or

tripolar spindles (cyan arrowheads). **C** Quantification of spindle defects observed through the first two cell divisions at 22.5 °C. The *zyg-1(it25); ssna-1(Δ)* double mutant embryos display twice as many monopolar spindles as *zyg-1(it25)* embryos. The double mutant also assembles multipolar spindles which are not observed in either single mutant. **D** Co-sedimentation assay showing SSNA-1 and ZYG-1 interact. Proteins were incubated alone (left) or in various combinations (right), separated into soluble (S) and pellet (P) fractions by centrifugation, and then detected by immunoblotting. Note that ZYG-1 is found in the soluble fraction when incubated alone or with microtubules but shifts to the pellet fraction upon incubation with SSNA-1. **E** Quantitation from three-independent co-sedimentation experiments. Mean and SD are shown. Source data are provided as a Source Data file.

19.3% of the double mutant embryos to be viable if the mutations did not interact (the product of the viabilities of the single mutants). Thus, we conclude that *zyg-1(it25)* and *ssna-1(Δ)* genetically interact, but in a negative manner.

We next performed live imaging of the same mutants expressing GFP::histone H2B and SPD-2::mCherry. Consistent with a weak block in centriole assembly, 14% of sperm-derived centrioles failed to duplicate in *zyg-1(it25)* embryos, leading to monopolar spindles at the second cell cycle (Fig. 5B, C). In contrast, the *ssna-1(Δ)* embryos produced only bipolar spindles through the two-cell stage, consistent with our prior observations (Fig. 5B, C). To our astonishment, the *zyg-1(it25); ssna-1(Δ)* double mutant showed an enhancement of both the monopolar and multipolar phenotypes; this strain exhibited about twice as many monopolar spindles as the *zyg-1(it25)* single mutant (27% vs 14%) while also exhibiting 23% multipolar spindles (Fig. 5B, C). Thus, the *zyg-1* mutation enhanced, rather than suppressed the multipolar spindle

phenotype of the *ssna-1(Δ)* mutation, indicating that the multipolar spindle phenotype arose through a mechanism other than over-duplication. Interestingly, the presence of both monopolar and multipolar spindles in the *zyg-1(it25); ssna-1(Δ)* double mutant appeared strikingly similar to the phenotype of *sas-1* mutant embryos where centriole stability is strongly impaired[27]. This suggests that a reduction of ZYG-1 activity in the *ssna-1(Δ)* mutant could result in structural failure during centriole assembly leading to monopolar spindle formation as well as structural failure post assembly leading to centriole fragmentation and multipolar spindle assembly (see Discussion).

To test if SSNA-1 and ZYG-1 physically interact, we performed an in vitro co-sedimentation assay with purified proteins. When incubated alone and subjected to centrifugation, ZYG-1 and BSA remained in the soluble fraction while filaments such as SSNA-1 and microtubules were found in the pellet to varying extents (Fig. 5D, E). However, when co-incubated with SSNA-1, ZYG-1 shifted mostly to the pellet fraction. No

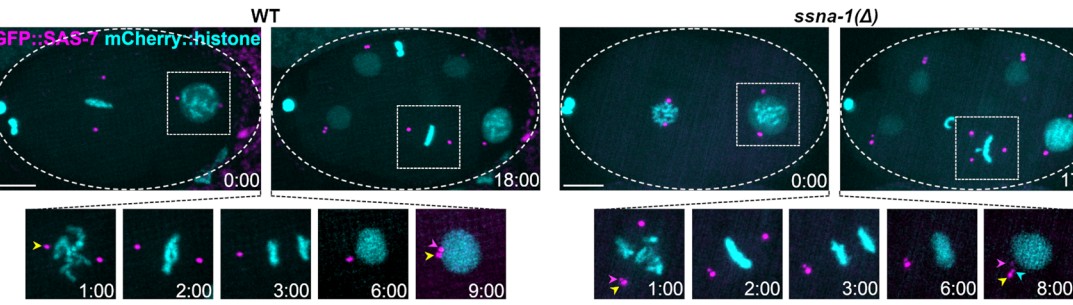

**Fig. 6 | Extra centrioles arise early during the cell cycle in *ssna-1*(Δ) embryos.** Time lapse images from wild-type or *ssna-1*(Δ) embryos expressing mCherry::histone and GFP::SAS-7. The first and last frames are shown at the top while enlargements show intervening time points with only one spindle pole shown beginning in anaphase (time point 3:00). In wild-type embryos two GFP::SAS-7 foci (corresponding to a disengaged centriole pair) first become visible at each spindle pole during telophase (*t* = 9:00, yellow and magenta arrowheads). During the ensuing cell cycle, this cell assembles a bipolar spindle (*t* = 18:00). In *ssna-1*(Δ) embryos, two GFP::SAS-7 foci first become apparent during prometaphase (*t* = 1.0, yellow and magenta arrowheads). During telophase when centriole separation normally occurs, a third GFP::SAS-7 spot appears (*t* = 8.0, cyan arrowhead). During the ensuring cell cycle this cell assembles a tripolar spindle (*t* = 17:00). Bar = 10 μm.

such shift was seen when ZYG-1 was incubated with microtubules or when BSA was incubated with SSNA-1, indicating that SSNA-1 and ZYG-1 specifically interact with each other. In summary, our results indicate that SSNA-1 and ZYG-1 genetically and physically interact and work together to ensure proper centriole number.

## SSNA-1 functions post-assembly to ensure the integrity of centrioles

To further investigate the origin of the extra centrioles in *ssna-1*(Δ) mutants, we imaged strains expressing GFP::SAS-7. Unlike the GFP::SPD-2 transgene which localizes to both centrioles and PCM (Fig. 4A), GFP::SAS-7 is strictly a centriole marker and thus provides a higher level of resolution. Wild-type embryos reproducibly exhibited a single focus of GFP::SAS-7 at each centrosome until telophase when the mother and daughter of each pair disengaged in preparation for the next round of duplication (Fig. 6 timepoint 9:00). Strikingly, in *ssna-1*(Δ) embryos, we could resolve two GFP::SAS-7 foci at some centrosomes as early as prophase (Fig. 6 timepoint 1:00). Later in telophase, we often detected a third focus that arose next to one of the two preexisting GFP::SAS-7 foci (Fig. 6 timepoint 8:00). Cells that inherited these centrosomes invariably assembled multipolar spindles during the next cell cycle (Fig. 6 timepoint 17:00). This indicates that the extra centrioles arise in close association with preexisting centrioles. This data is consistent with either of two defects: premature mother daughter disengagement leading to a second round of duplication in the same cell cycle (reduplication)[58,59] or structural failure leading to centriole fragmentation.

To discriminate between these two possibilities, we developed an assay to directly observe occurrences of premature centriole disengagement and centriole fragmentation (Fig. 7A). In this assay we mated males carrying an endogenous *sas-4* gene tagged with the SPOT epitope (magenta) to hermaphrodites expressing an endogenously tagged *sas-4::gfp* gene (green). Because SAS-4 is a stable component of centrioles[24], we were able to follow the fate of magenta sperm centrioles as well as determine the time of disengagement of their green daughters. In wild-type embryos we would expect to always find exactly two magenta sperm-derived centrioles and show that these magenta mother centrioles and their green daughters are coincident until disengagement in anaphase. In the mutant, centriole fragmentation would give rise to more than two magenta centrioles while premature disengagement would present as loss of mother daughter coincidence before anaphase. As shown in Fig. 7B, C, while meiotic stage wild-type embryos contained a single SPOT::SAS-4 focus representing the closely apposed sperm centriole pair, following their disengagement in early first prophase, we always detected two SPOT-tagged centrioles in older one- and two-cell embryos. Further, in wild-

type embryos mother and daughter centrioles were always engaged prior to mid-anaphase as expected (Fig. S5A–D; *n* = 62 centrosomes). A different outcome prevailed in *ssna-1*(Δ) embryos. While all prophase stage one-cell mutant embryos contained two SPOT-tagged foci indicating that *ssna-1*(Δ) sperm donated the correct number of centrioles, we frequently observed more than two male derived SPOT-tagged centrioles in older one- and two-cell embryos, demonstrating that the sperm-derived centrioles subsequently lost structural integrity and fractured (Figs. 7B, C and S5A). To confirm this, we measured the relative intensities of SPOT::SAS-4 labeled sperm centrioles and found that in *ssna-1*(Δ) embryos where three sperm-derived centrioles were present, one of the three was always much brighter than the other two (Fig. 7D). In contrast, in wild-type embryos the two sperm-derived centrioles exhibited intensities that were more similar. This result is most consistent with the extra centrioles arising from breakage of one of the two sperm-derived centrioles leading to two dim fragments and a relatively bright intact centriole.

In the fragmentation assay we also observed instances where blastomeres contained one magenta and two green foci which likely occurred when those centrioles that had formed in the embryo had fragmented (Fig. S5B). Importantly, we did not detect any instances of premature disengagement in the *ssna-1*(Δ) mutant (Fig. S5A–D; *i* = 56 centrosomes). Even in instances where we observed two magenta centriole fragments prior to late anaphase, one of the two fragments was always associated with a green daughter indicating that disengagement had not yet occurred (Fig. S5A). We conclude that the extra centrioles in *ssna-1*(Δ) mutants arise through fragmentation.

In light of our results showing that sperm-derived centrioles lacking SSNA-1 fracture following fertilization of *ssna-1*(Δ) eggs (Fig. 7B, C), we found it intriguing that we did not detect an embryonic lethal phenotype when *ssna-1*(Δ) mutant males were mated to wild-type females (Fig. 1G). Since the sperm-derived centrioles in these experiments lacked SSNA-1, we might have anticipated them to fragment, leading to multipolar spindle formation and embryonic lethality. To investigate this, we repeated the fragmentation assay, this time mating *ssna-1*(Δ) mutant males to wild-type hermaphrodites. In contrast to the loss of structural integrity observed when *ssna-1*(Δ) centrioles are introduced into *ssna-1*(Δ) oocytes, *ssna-1*(Δ) centrioles maintained their structural integrity when introduced into wild-type oocytes (Fig. 7B, C). This indicates that cytoplasmic SSNA-1 present in the egg can be recruited to the fully formed sperm centrioles to stabilize them after fertilization. Indeed, we found this was the case, as sperm centrioles donated by *ssna-1*(Δ) males became enriched for maternally supplied SSNA1::C-tag as early as meiosis following fertilization (Fig. S5E).

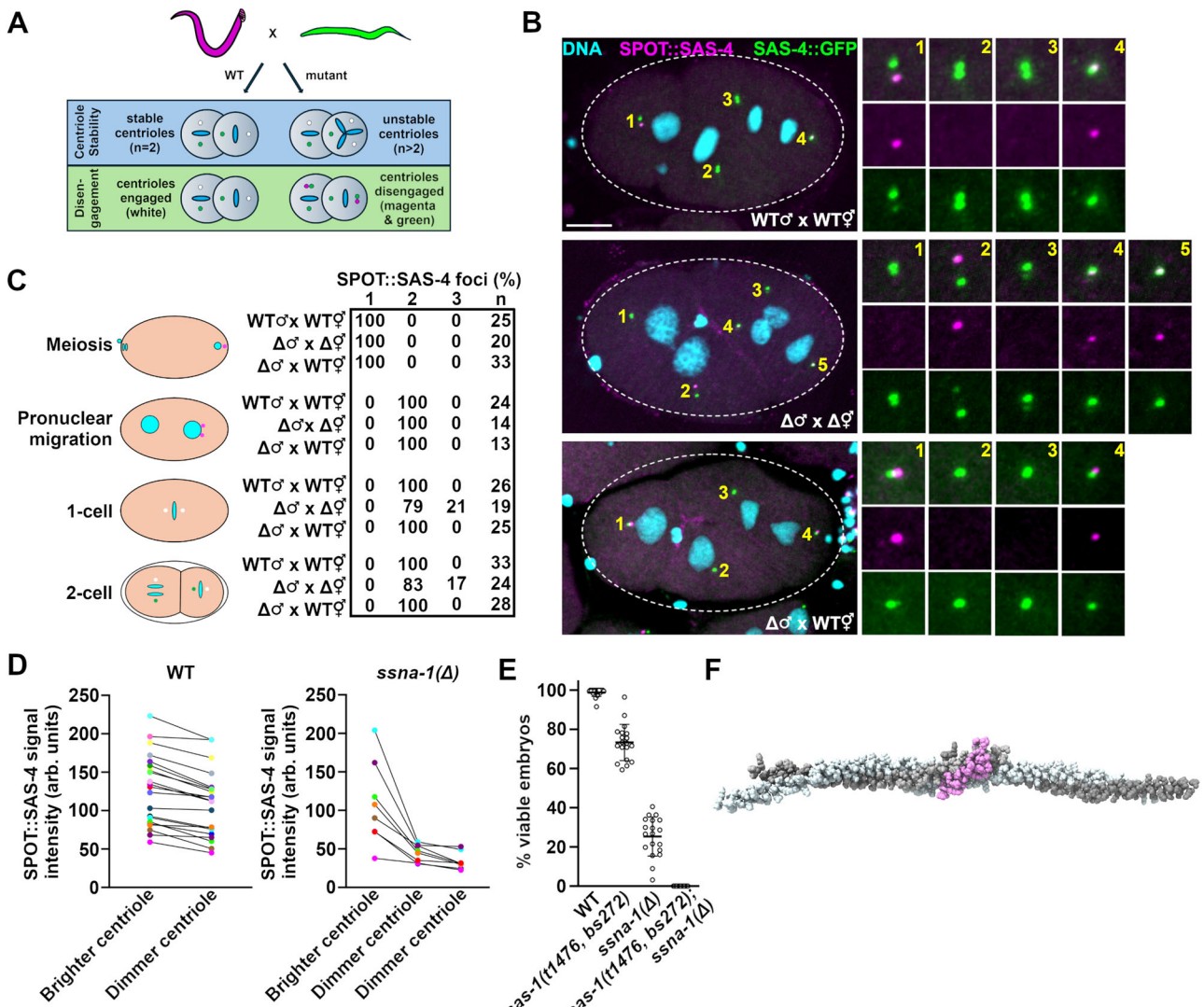

**Fig. 7 | SSNA-1 is required for the structural integrity of centrioles. A** Assay to detect centriole fragmentation and premature disengagement. Males whose centrioles are marked with SPOT::SAS-4 (magenta) are mated to hermaphrodites expressing SAS-4::GFP (green). In wild-type embryos the two paternal centrioles will indelibly be marked magenta while new centrioles will be marked green. Magenta mother/green daughter centriole pairs will remain engaged (and appear white) until telophase. Centriole instability will be detected as >2 magenta centrioles per embryo while premature disengagement will present as loss of green-red coincidence prior to telophase. **B** WT × WT (top), *ssna-1(Δ)* × *ssna-1(Δ)* (middle), and *ssna-1(Δ)* × WT two-cell stage embryos stained for SPOT::SAS-4 (magenta), SAS-4::GFP (green), and DNA (cyan). The WT × WT and *ssna-1(Δ)* × WT embryos possess two magenta centrioles indicating centriole stability while the *ssna-1(Δ)* embryo possesses three magenta centrioles revealing a centriole fragmentation phenotype. Note that in the wild-type embryo, one of the blastomeres in telophase is undergoing centriole duplication where one of the sperm centrioles (1) is producing a

new green daughter. Bar = 10 μm. **C** Quantitation of centriole fragmentation in wild-type and *ssna-1(Δ)* mutant embryos. Note that in early prophase *ssna-1(Δ)* mutant embryos have two sperm derived centrioles indicating *ssna-1(Δ)* sperm contain the normal number of centrioles. However, older *ssna-1(Δ)* embryos frequently contain more than two magenta foci indicating that the sperm centrioles have fragmented. **D** Quantitation of SPOT::SAS-4 intensities. Lines connect centrioles from the same embryo. In *ssna-1(Δ)* embryos with three sperm derived centrioles, one is always significantly brighter than the other two, while in wild-type embryos, the sperm centrioles are much more similar in intensity. **E** *ssna-1* and *sas-1* genetically interact. The percentage of viable embryos produced by wild-type *n* = 23, *ssna-1(Δ)* *n* = 19, *sas-1(t1476, bs272)* *n* = 22, and *sas-1(t1476, bs272); ssna-1(Δ)* *n* = 21 hermaphrodites. Each datapoint represents the progeny of a single hermaphrodite. Mean and SD are shown. **F** AlphaFold prediction of SSNA-1 tetramer (grey) and SAS-1 (a.a. 539–570) (pink) showing an interaction with a pTM score of 0.65. The SSNA-1 tetramer structure is from Agostini et al.[36] Source data are provided as a Source Data file.

As SSNA-1 and SAS-1 are both required for centriole stability, localize in close proximity, and exhibit similar mutant phenotypes, we tested for a genetic interaction. We constructed a *sas-1(t1476, bs272); ssna-1(Δ)* double mutant and compared its viability to each of the single mutants. As shown in Fig. 7E, at 20 °C, the *sas-1(t1476, bs272)* strain exhibits 73% embryonic viability while the *ssna-1(Δ)* strain exhibits 25% viability. In contrast the *sas-1(t1476, bs272); ssna-1(Δ)* double mutant was fully inviable. The observed level of viability was significantly lower than what would be expected if the two genes failed to interact ($0.73 \times 0.25 = 0.18$ or 18% viability). Thus *ssna-1* and *sas-1*

interact, indicating that they function together to establish centriole stability. We then used AlphaFold modeling to determine if the two proteins are likely to physically interact and found that there was a high probability that the C-terminal residues of SAS-1 could dock onto the triple stranded junction of SSNA-1 (Fig. 7F). Finally, we investigated if loss of SSNA-1 affects the localization of SAS-1 to centrioles. As shown in Fig. S5F, the centriole level of endogenously tagged SPOT::SAS-1, as well as another centriole marker (GFP::SAS-7), was markedly reduced in the *ssna-1(Δ)* mutant relative to the wild type, indicating that loss of SAS-1 contributes to the fragility of *ssna-1(Δ)* centrioles. Interestingly,

we found that SAS-1 also localizes to centriole satellite-like structures and does so in a SSNA-1-dependent manner (Fig. S5F), underscoring the close association between these two centriole-stabilizing factors.

## Discussion

Centriole amplification is a common feature of cancer cells and has generally been thought to occur as a result of excess centriole formation in response to either elevated levels of centriole assembly factors such as PLK4, STIL, or SAS-6[60] or abnormal centriole elongation that results in the formation of ectopic daughter centrioles[12,61,62]. However, it has been shown that amplification can also occur directly through the fragmentation of existing centrioles[12]. Despite this, the underlying causes of centriole fragmentation have not been thoroughly explored. Here we have identified SSNA-1 as a centriole stability protein. SSNA-1 is critical for the structural integrity of centrioles, rather than centriole assembly. In *ssna-1* null mutant embryos, centrioles undergo fragmentation leading to centriole amplification, multipolar spindle formation, and abnormal cell division.

While our results clearly demonstrate that many of the extra centrioles in *ssna-1* null mutants arise through fragmentation of existing centrioles, we cannot completely rule out the possibility that some of the extra centrioles are produced by a reduplication mechanism. Normally, a second round of duplication during a single cell cycle is prevented by a licensing mechanism, whereby engaged mother-daughter pairs are unable to initiate duplication until they disengage from one another during the ensuing cell cycle, thereby becoming licensed for a new round of duplication[58,63,64]. While our data indicates that mother-daughter pairs are not prematurely disengaging in the *ssna-1* mutant (Figs. 7 and S5), it is possible that reduplication occurs as a result of centriole breakage, where upon a centriole fragment lacking an associated daughter could be produced. This fragment might then act as a licensed mother to initiate a second round of duplication within a single cell cycle. Such events would not have been readily detectable in our fragmentation/premature disengagement assay. Thus, the possibility exists that fragmentation could lead to centriole amplification through two mechanisms: fragmentation itself and fragmentation followed by reduplication.

Our data shows that SSNA-1 is a stable component of centrioles that localizes in the vicinity of the central tube where the centriole stability factor SAS-1 resides. Interestingly, most of the EM densities within the *C. elegans* centriole have been assigned to one of the known centriole proteins[7]. However, a single density on the side of each microtubule opposite SAS-1 remains unassigned. It is possible that SSNA-1 is the missing protein and that it functions in conjunction with SAS-1 to stabilize the structure. Consistent with this idea, *ssna-1* and *sas-1* mutations display a strong genetic interaction (Fig. 7E). Despite this finding, *ssna-1* and *sas-1* mutants do not exhibit identical phenotypes. A complete knockout of *ssna-1* is still partially viable at 25 °C (Fig. 1E) whereas *sas-1* hypomorphic mutations are fully inviable at this temperature[27]. Further, *ssna-1* null mutants only display multipolar spindles and these almost always form after the two-cell stage. In contrast, *sas-1* mutant embryos show defects beginning at the earliest stages of embryogenesis and assemble both monopolar and multipolar spindles[27]. The more severe phenotypes of *sas-1* mutants compared to *ssna-1* mutants likely reflect differences in the roles played by the two proteins (Fig. 8). While SAS-1 is thought to stabilize the centriole both during and after its assembly[27], our data indicates that SSNA-1 is only required to stabilize the centriole following assembly.

Unexpectedly, our study reveals that a partial loss of ZYG-1 can markedly enhance the phenotype of *ssna-1* mutants, suggesting that ZYG-1 plays a role in centriole stability. So far, the only known role of ZYG-1 is to initiate centriole assembly by binding and recruiting the SAS-5-SAS-6 complex[18] and by phosphorylating SAS-5 to stably incorporate the SAS-5-SAS-6 complexes that form the cartwheel[22]. Since the cartwheel has been shown to stabilize vertebrate procentrioles[2], it

seems plausible that the ability of a *zyg-1* hypomorphic allele to enhance the *ssna-1* instability defect is due to the loss of ZYG-1's role in assembly rather than some distinct post-assembly function. That is, less ZYG-1 activity could lead to a reduction in the amount of stably associated SAS-5-SAS-6 complexes; this would further destabilize centrioles lacking SSNA-1. However, it is still possible that ZYG-1 might function to stabilize centrioles independently of its role in assembly. Consistent with this idea, ZYG-1 is found at centrioles throughout the cell cycle and not just when centriole assembly is occurring[65]. Furthermore, we have found that ZYG-1 and SSNA-1 physically interact (Fig. 5D, E) suggesting that ZYG-1 might regulate SSNA-1 mediated centriole stability. Further study will be needed to determine precisely how ZYG-1 contributes to the structural integrity of centrioles.

We propose a model to explain how the various structural elements of the centriole contribute to its stability (Fig. 8). In this model, loss of structural integrity during centriole assembly leads to loss of the daughter and ultimately monopolar spindle formation, while loss of structural integrity following assembly leads to centriole fragmentation and multipolar spindle formation (Fig. 8). SAS-1 is clearly needed for both processes. In contrast, while SSNA-1 contributes to stability both during and after assembly, it is only required post assembly. However, SSNA-1's contribution to stabilizing the centriole during assembly can be revealed if cartwheel assembly is perturbed, as occurs when ZYG-1 is partially inhibited in the *ssna-1(Δ)* mutant. Thus, the cartwheel, SSNA-1, and SAS-1 all contribute to maintaining the structural integrity of centrioles.

Our findings are likely relevant to centriole stability in vertebrates. SSNA1 and C2CD3, the human homologs of SSNA-1 and SAS-1 respectively, both localize to centrioles with C2CD3 being restricted to the distal end[28,30,32,34,66]. Similar to SAS-1, ectopically expressed C2CD3 appears to stabilize cytoplasmic microtubules and in its absence, centrioles are shorter with abnormal distal architecture[27–30]. Likewise, human SSNA1 has been shown to directly stabilize microtubules in vitro where it modulates dynamic instability, binds to sites of microtubule damage, and counteracts the microtubule severing activity of spastin[35]. Thus it seems possible that SSNA-1 and C2CD3 function together at centrioles where they could stabilize the microtubule blades of the outer wall.

Another interesting aspect of our study is the apparent tissue-specific role of SSNA-1. While SSNA-1 seems to be present on all centrioles, we only observe a major effect during embryogenesis. So why does loss of SSNA-1 not have obvious effects in the maternal or paternal germ line or in the soma? There are a few of possible explanations for the apparent lack of defects outside of embryogenesis. One possibility is that centriole fragmentation is happening in most tissues but that it is easiest to detect in embryos because they are exquisitely sensitive to failures in cell division. Even just a single failed division during the earliest stages of embryogenesis would likely result in lethality[67]. Other tissues may not be so sensitive and could experience a similar frequency of centriole fragmentation and aberrant cell divisions while still developing normally. Alternatively, it is possible that fragmentation is really occurring at a significantly higher rate in the embryo than in other tissues. This might be because other proteins function redundantly with SSNA-1 outside of embryogenesis or that centrioles experience particularly high levels of mechanical stress during the early embryonic divisions when relatively large spindles are being actively positioned due to microtubule-dependent forces operating at this time. Indeed, our observation that centriole fragments first become visible during the late second cell cycle (Fig. 6) and not earlier, is consistent with a model in which centrioles lacking SSNA-1 gradually weaken in response to the cumulative damage caused by mechanical stress. In depth cytological analysis will be required to figure out which of these possibilities might be true.

While our work establishes a role for SSNA-1 at centrioles, what it might be doing at satellite-like structures remains an open question.

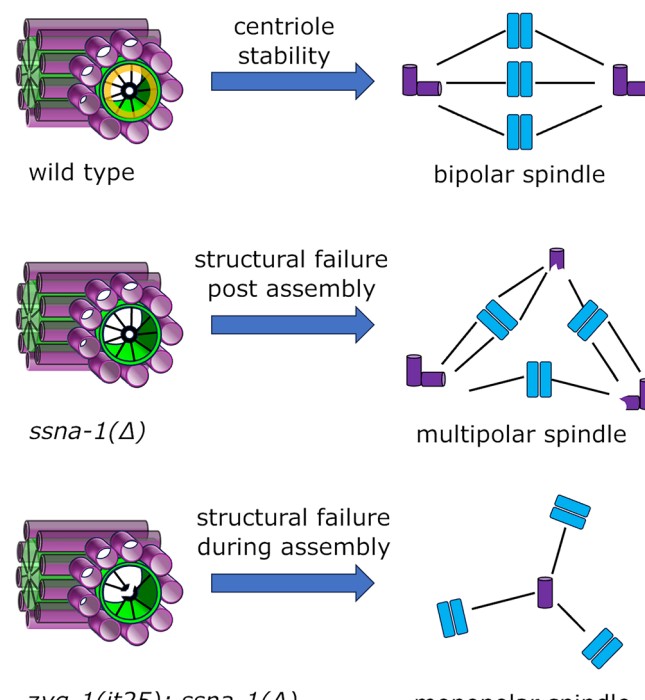

**Fig. 8 | SSNA-1 functions to stabilize centrioles during and after assembly.** A model depicting the fate of centrioles possessing or lacking stabilizing elements. Top. Wild-type centrioles, which possess both SSNA-1 (yellow ring) and a fully formed cartwheel, exhibit long-term stability resulting in faithful bipolar spindle assembly. Middle. Centrioles lacking SSNA-1 can be assembled but lack long-term stability leading to centriole fragmentation and multipolar spindle formation. Bottom. Centrioles lacking SSNA-1 and a fully formed cartwheel (due to partial inhibition of *zyg-1*) experience structural failure during assembly leading to loss of the daughter centriole and monopolar spindle assembly. Microtubules are in purple, SSNA-1 is in yellow, and the cartwheel is in black. The green cylinder highlights the lumen of the centriole.

Our analysis of the SSNA-1$^{R18E}$ mutant suggests that the pool of SSNA-1 localized to satellite-like structures does not have an essential role in the embryo. In vertebrates, centriole satellites have been implicated in the storage and trafficking of centriole proteins[54] and are composed of hundreds of proteins[55,56]. Prominent among these is a large coiled-coil protein called pericentriolar material 1 (PCM1) which is thought to form a structural scaffold, as depletion of PCM1 results in loss of satellites[68]. A putative *C. elegans* PCM1 ortholog has been identified[37] but has not yet been characterized, leaving unanswered the question of how similar satellite-like structures are to centriolar satellites. However, C2CD3 has been found to localize to centriolar satellites[29] suggesting both structures are enriched for centriole stability factors.

In summary, our work identifies SSNA-1, a conserved filamentous microtubule-binding protein as an important structural component of centrioles. SSNA-1, like SAS-1, localizes between the cartwheel and the microtubules of the outer wall where it functions to stabilize the centriole. Centrioles can be assembled without SSNA-1, but fragment giving rise to multipolar spindles. In contrast, if SSNA-1 is absent and cartwheel assembly is inefficient, centrioles fail to form, giving rise to monopolar spindles. Thus by ensuring the long-term persistence of centrioles, SSNA-1 contributes to the numerical control of centrioles.

## Methods

### Worm strains and maintenance
Worms were maintained at 20 °C on MYOB agar plates seeded with *Escherichia Coli* OP50 according to standard protocols[69]. All *C. elegans* strains used in this study are listed in Table S1.

### CRISPR-Cas9 genome editing
CRISPR-Cas9 genome editing was performed as previously described[70,71]. Primers and oligonucleotide repair templates were purchased from Integrated DNA Technologies (Coralville, IA) and crRNAs were purchased from Dharmacon, Inc. (Lafayette, CO). Cas9 protein was purchased from PNA Bio, Inc. All crRNA and repair template sequences are listed in Table S2.

### Embryonic viability assays
For quantification of embryonic viability, L4 larvae were picked individually to 35 mm MYOB plates and incubated at the indicated temperature for approximately 24 h. Worms were then transferred to new 35 mm plates and allowed to lay eggs for 24 h after which the adult worms were removed. The plates were returned to the incubator for an additional 24 h before dead eggs and viable larvae were scored.

### Fixed and live imaging
Immunofluorescence microscopy was performed as previously described[72]. Mouse monoclonal anti-alpha-tubulin DM1A (cat# CP06, Sigma-Aldrich, St. Louse, MO), affinity-purified rabbit polyclonal anti-SAS-4 peptide (MASDENIGADGEQKPSC-amide) antibody (Yenzym Antibodies, Brisbane, CA), and affinity-purified rabbit polyclonal anti-SSNA-1 raised against the full purified protein (Yenzym Antibodies, Brisbane, CA) were used at a dilution of 1:1000. Alexa Fluor 488 anti-rabbit (cat# A32731) and Alexa Fluor 568 anti-mouse (cat# A-11004) secondary antibodies (Thermo Fisher Scientific, Waltham, MA) were used at a 1:1000 dilution. The SPOT tag was detected using the SPOT-Label Alexa Fluor 568 (cat# ebAF568, Proteintech, Rosemont, IL) and the C-tag was detected using the CaptureSelect Alexa Fluor 488 anti-C-tag secondary (cat# 7213252100, Thermo Fisher Scientific, Waltham MA), each at a dilution of 1:1000. The anti-SPOT and anti-C-tag nanobodies were incubated at room temperature for 2 h while all other antibodies were incubated at 4 °C overnight. For dual labeling, nanobody incubation occurred following secondary antibody incubation.

For time-lapse imaging of embryos, a circular silicone gasket of .25 mm thick × 16 mm diameter with an 8 mm center circular well was created with a hollow punch using CultureWell$^{Tm}$ Sheet Material (cat # CWS-S-0.25, Grace Bio-Labs, Inc. Bend, OR) and then placed onto a glass slide. The well was then filled with 18 μL of molten 3% agarose in Egg buffer (188 mM NaCl, 48 mM KCl, 2 mM CaCl$_2$, 2 mM MgCl$_2$, 25 mM HEPES, pH 7.3) and covered with a second glass slide to create an agar pad. L4 larvae were shifted to the indicated temperature the day prior to imaging. Gravid adults were then dissected into 1 μL Egg buffer on a 12 mM circular #1 coverslip. The slide containing the gasket and agar pad was then gently inverted onto the coverslip to create an airtight seal and the embryos were imaged at the temperatures indicated above.

Spinning disk confocal microscopy of live and fixed specimens employed a Nikon Eclipse Ti microscope equipped with a Plan Apo VC 60 × 1.4 N.A. oil immersion lens, a CSU-X confocal scanning unit (Yokogawa Electric Corporation, Tokyo, Japan), and an Orca-Fusion BT C15440 digital camera (Hamamatsu Photonics, Shizuoka, Japan). For temperature control, a Thermo Plate heating/cooling stage (Tokai Hit USA Inc, Bala Cynwyd, PA) was used. Excitation light was generated using 405, 488, and 561 nm solid state lasers controlled via a Nikon LunF-XL laser module (Nikon Instruments, Inc, Tokyo, Japan). Images were acquired using NIS-elements software (Nikon Instruments, Inc, Tokyo, Japan).

### Image analysis and intensity quantification
Image processing was performed using either Image J2 version 2.14.0/ 1.54 f or NIS-elements software (Nikon Instruments, Inc, Tokyo, Japan). For total intensity measurements in Fig. S4, maximum intensity projections of a 3.75 μm (x) × 3.75 μm (y) × 4 μm (z) volume centered around the centriole or a 7.5 μm (x) × 7.5 (y) μm × 10 μm (z) centered

around the PCM were created and processed using Otsu thresholding. Integrated density measurements were then taken for each projected image. For intensity measurements in Figs. 7, S2, and S5, a substack of 7.5 μm (x) × 7.5 (y) μm was created around each centrosome, followed by a maximum intensity projection surrounding the centrosome. Background signal was subtracted using the Subtract background tool with a rolling ball radius of 3 pixels. A 1.25 μm circular ROI was then used to measure the integrated density for each centrosome.

## Ultrastructure expansion microscopy (U-ExM)

Embryos and gonads were expanded using a modified version of the U-ExM protocol as previously described[50,51,73]. Briefly, slides were coated with 8 μL of 0.1% poly-L-lysine (cat # P8920, MilliporeSigma, Burlington, MA), evenly spread with a pipet tip, and briefly heated for 4 and 5 s at 100 °C (until the droplet is dry) just before usage. To clean worms, 10 μL of a mixture composed of 50% water and 50% M9 buffer (supplemented with 1 mM of MgSO4 in filtered water) was pipetted in the center of a non-coated slide and 20–30 young adult hermaphrodites were deposited in the drop. Using an eyelash, worms were transferred to 20 μL of the same solution on a coated slide and concentrated in the center to facilitate subsequent steps. The worms were cut open with a scalpel to release the embryos or gonads, and the carcasses removed using an eyelash. After placing a 15-mm × 15-mm square coverslip on top of the specimen, the excess liquid was carefully and quickly removed using a piece of filter paper. The removal was halted as soon as a few embryos began to rupture. The slide was then quickly plunged into liquid nitrogen, the coverslips flicked off with a razor blade, and the slides immediately plunged into precooled −20 °C methanol for 5 min at −20 °C. The slides were then immersed twice in PBS for 5 min each. A round spacer of 0.3 mm thickness (cat # IS317, SunJin Lab Co) was placed on each slide surrounding the specimen and 300 μL of anchoring solution (2% acrylamide and 1.4% formaldehyde diluted in PBS) was added to the well created by the spacer. The slides were incubated overnight at 37 °C.

The next day, the monomer solution (19% sodium acrylate, 0.1% bis-acrylamide, and 10% acrylamide) was thawed on ice. The anchoring solution was removed, and 90 μL of monomer solution (without TEMED and APS) was added on top of the specimen. The slides were incubated for 15 min on a chilled metallic block on ice before the spacers were removed with tweezers. Using a hydrophobic PAP PEN marker, $a$ - 12 mm circle was drawn around the sample and monomer solution containing 0.5% each of TEMED and APS was placed on top of the sample (35 μL final volume). The sample was covered with a 12-mm round coverslip to initiate gelation. The slides were incubated for 15 min on the cold metallic block and then 1 h at 37 °C in a humidity chamber. After gelation, slides were submerged in denaturation buffer (200 mM SDS, 200 mM NaCl, and 50 mM Tris in nuclease-free water, pH 9) and placed on a shaker (110 rpm) at room temperature for 15–30 min to let the gels detach from the slides and coverslips. The gels were then carefully transferred to individual 1.5-ml microcentrifuge tubes containing 1 mL of denaturation buffer. Denaturation was performed for 1.5 h at 95 °C followed by 3 washes for 30 min each in filtered H₂O. The gels were then measured with a ruler to calculate the expansion factor by dividing the final size of the gel after expansion by the initial dimension of 12 mm. Next, the edges of the expanded gels were cut with a razor blade to fit inside a 6-well plate. The gels were shrunk using three exchanges of PBS for 10 min each to fit within a single well of a 12-well plate. Gels were then stained using 1:250 dilutions of the following primary antibodies in PBS-BSA 2% overnight at 4 °C under mild agitation (80 rpm): rabbit anti-GFP antibody (cat# TP401, Torrey Pines Biolabs), rat anti-HA (cat# 11867423001, MilliporeSigma, Burlington, MA), rabbit anti-SSNA-1 (this study), mouse anti-FLAG (cat# F1804, MilliporeSigma, Burlington, MA) and guinea pig anti-alpha and beta tubulin antibodies (cat#

ABCD_AA345 and cat# ABCD_AA344, ABCD antibodies, Geneva Switzerland).

The next day, the gels were washed in PBS with 0.1% Tween three times at room temperature with agitation (110 rpm), and then incubated with 1:400 dilutions of the following secondary antibodies in PBS-BSA 2% for 2.5 h at 37 °C under agitation (110 rpm): Alexa Fluor 488 anti-rabbit (cat# A11008), Alexa Fluor 568 anti-mouse (cat # A11004), Alexa Fluor 488 anti-rat (cat# A21208), Alexa Fluor 568 anti-guinea pig (cat # A11075, Thermo Fisher Scientific, Waltham, MA) or anti-guinea pig Cy5 (cat # 706-175-148, Jackson ImmunoResearch, West Grove, PA) and DAPI (cat# A1001, BioChemica). Finally, gels were washed three times with agitation in PBS with 0.1% Tween at room temperature and then fully expanded with three 10 min washes in filtered H₂O in a Petri Dish.

For image acquisition and analysis, expanded gels were cut with a razor blade into squares to fit into a 36 mm metallic imaging chamber. Excess water was carefully removed, and the gel was mounted onto 24 mm coverslips coated with 0.1% poly-D-lysine to prevent drifting. Images were taken with an inverted confocal Leica TCS SP8 microscope using a 63 × 1.4 NA oil immersion objective. 3D stacks were acquired with 0.12 μm z-intervals and a 35–45 nm x, y pixel size. The images were then deconvolved using a lightning mode at max resolution, adaptive as "Strategy" and water as "Mountain medium". The diameter quantifications were performed as previously published[3].

## Quantitative immunoblotting

Quantitative western blotting was performed as previously described[53]. Briefly, gravid hermaphrodites were transferred to tubes containing M9 (22 mM KH₂PO₄, 22 mM Na₂HPO₄, 85 mM NaCl, 1 mM MgSO₄), washed three times with M9, and then suspended in homemade 4× LDS-NuPAGE sample buffer (549 mM Tris base, 432 mM Tris-HCL, 2.05 mM EDTA, 8% lithium dodecyl sulfate, 40% glycerol, 4.4 mM Orange G) to a final concentration of 1× and then boiled at 95 °C for 10 min. Samples were then sonicated in a BioRuptor (Diagenode, Liège, Belgium). Proteins from the equivalent of 50 gravid adults from each sample was then run on a NuPAGE 4–12% Bis-Tris precast gel (Thermo Fisher Scientific, Waltham MA) followed by transferring to a nitrocellulose membrane using the iBlot semi-dry transfer system (Thermo Fisher Scientific, Waltham, MA). Membranes were blocked in Odyssey blocking buffer (PBS) (LiCOR Biosciences, Lincoln, NE) and then probed with the following primary antibodies at a 1:1000 dilution: rabbit anti-SAS-5[53], guinea pig anti-SAS-6[55], and mouse anti-α-tubulin DM1A (cat# CP06, MilliporeSigma, Burlington, MA). Anti-mouse 680 (cat# 926-68070), anti-rabbit 800 (cat # 926-32211), and anti-guinea pig 800 (cat# 926-32411) IRDye secondary antibodies (LiCOR Biosciences, Lincoln, NE) were used as a 1:14,000 dilution. Membranes were imaged on an Invitrogen iBright 1500 (Thermo Fisher Scientific, Waltham, MA). Band intensity quantitation was measured using ImageJ2 version 2.14.0/1.54f software and normalized to the α-tubulin loading control.

## Assay for premature disengagement and centriole fragmentation

To assay for centriole premature disengagement and centriole fragmentation *spot::sas-4 or spot::sas-4; ssna-1(Δ)* males were mated to *sas-4::GFP* or *sas-4::GFP; ssna-1(Δ)* L4 hermaphrodite larvae as indicated. Matings were carried out on 35 mm OP50-seeded MYOB plates in a 2:1 ratio (40 males:20 hermaphrodites) for ~48 h. Gravid hermaphrodites were then dissected on poly-L-lysine coated slides, freeze cracked in liquid nitrogen, and subjected to immunofluorescence staining as described above with the SPOT-Label Alexa Fluor 568 (cat# ebAF568, Proteintech, Rosemont, IL) for 2 h at room temperature. Native GFP fluorescence was used to identify SAS-4::GFP. Embryos were then analyzed as described above and the number of SPOT::SAS-4 foci were quantified per embryo. Mating of *spot::sas-4; ssna-1(Δ)* males to *ssna-1::C-tag* hermaphrodites was

performed as described above but were carried out overnight. Immunofluorescence staining was performed as described above with SPOT and C-tag nanobodies.

## MT binding and sedimentation assay

SSNA-1 was assessed for its binding to ZYG-1 through high-speed centrifugation of purified proteins (in vitro reconstitution). SSNA-1 was dialyzed against resuspension buffer (50 mM phosphate buffer/NaOH pH 7.5, 150 mM NaCl, 10% glycerol (v/v), 1 mM DTT) overnight at 4 °C to grow filaments, and successively incubated at room temperature for 10 min with ZYG-1. The final reaction had a total volume of 30 μL with concentrations of 1.25 μM and 10 μM for ZYG-1 and SSNA-1, respectively. The sedimentation was performed at 80,000 rpm ($278,088 \times g$) with the rotor TLA-120.2 (Beckman Coulter Life Sciences, Brea, CA) for 10 min at 25 °C. Similarly, we tested the binding of ZYG-1 and pre-polymerized microtubules (MTs), as well as the binding between SSNA-1 and BSA. Brain tubulin was pre-polymerized for 30 min at 37 °C in BRB80 buffer (PIPES/KOH 10 mM pH 6.8, 1 mM $MgCl_2$, and 1 mM EGTA) supplemented with 1 mM GMPCPP. While the final concentrations were as follows: 1.25 μM each for ZYG-1 and BSA, and 10 μM each for SSNA-1 and MTs. Each experiment was performed in triplicate. Supernatants and pellets were then recovered and analyzed by SDS-PAGE. Gels were stained with Coomassie and oriole solutions, as well as imaged by immunoblotting as described above using mouse monoclonal anti-FLAG M2 (cat# F1804, MilliporeSigma, Burlington, MA), mouse monoclonal anti-α-tubulin DM1A, and rabbit polyclonal anti-SSNA-1 for ZYG-1, α−tubulin, and SSNA-1 detection, respectively. Band intensity quantification was performed using ImageJ2 version 2.14.0/1.54f.

Additionally, SSNA-1 was also assessed for its binding to MTs through low-speed centrifugation. Brain tubulin was pre-polymerized as previously described. SSNA-1 constructs were prepared in BRB80 buffer and pre-sedimented twice at $16,100 \times g$ using rotor TLA-120.2 for 10 and 5 min, respectively, at 25 °C to remove large oligomers. Soluble SSNA-1 was successively incubated for 4 min at room temperature with pre-polymerized MTs. The final reaction had a total volume of 30 μL with a concentration of 20 μM for each protein, and were sedimented at 15,000 rpm ($9,776.5 \times g$) using rotor TLA-120.2 for 5 min at 25 °C. Each experiment was performed in triplicate. The supernatants and pellets were then recovered and analyzed by SDS-PAGE. The imaging of the gels was carried out through Coomassie staining. Band intensity quantification was performed using Image J2 version 2.14.0/1.54f.

## Statistics

The generation of scatter plots and bar graphs were created using Prism 10 (GraphPad Software, San Diego, CA). For statistical analysis, we performed unpaired two-tailed Student's $t$ test, and one-way ANOVA with Tukey's multiple comparisons test in Prism. Figures show only relevant and statistically significant comparisons. Further information is provided in figure legends.

## Reporting summary

Further information on research design is available in the Nature Portfolio Reporting Summary linked to this article.

## Data availability

Source data for quantitative analyses and westerns are available in the figshare https://doi.org/10.6084/m9.figshare.28592591 repository. Source data are provided with this paper.

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

## Acknowledgements

We thank Yan Lui, Lindsey Gentry, Valentin Burdet, and Florian Steiner for technical help and fruitful discussions. We also thank Pierre Gönczy for generously providing worm strains. Research within the KOC lab was supported by a grant (DK024962) from the Intramural Research Program of The National Institute of Diabetes and Digestive and Kidney Diseases (NIDDK) of the National Institutes of Health. N.M. was supported by the Division of Intramural Research Program of the National Heart, Lung, and Blood Institute (1ZIAHL006264) and the National Institute of Arthritis and Musculoskeletal and Skin Diseases of the National Institutes of Health. P.G. and V.H. were supported by the Swiss National Foundation SNSF 310030_205087.

## Author contributions

J.A.P. conceived the study, conducted experiments, analyzed data and prepared text and figures. L.A. and L.B. performed experiments, analyzed data and prepared the text. A.P., P.S., and Z.G.B. conducted experiments. V.H. and P.G. supervised experiments. C.B. and N.M. supervised experiments, analyzed data, and prepared images. K.F.O. supervised the work, analyzed data, and prepared text and figures.

## Funding

## Competing interests

The authors declare no competing interests.
