## [Peer Review file · Nature Communications]

***C. elegans* SSNA-1 is required for the structural integrity of centrioles and bipolar spindle assembly**

Corresponding Author: Dr Kevin O'Connell

Version 0:

Reviewer comments:

Reviewer #1

(Remarks to the Author)

Here Pfister et al explore the role of SSNA-1, the *C. elegans* ortholog of Sjogren's Syndrome Nuclear Antigen 1. This protein had been previously localized to the centrosome and basal body in other systems, but its cellular function had not been identified. Through endogenous labeling studies, the authors demonstrate that SSNA-1 is a stable component of the centriole in *C. elegans*, localizing to the inside of the centriole barrel through expansion microscopy as well as to centrosome satellite-like structures that appear near centrosomes at the end of mitosis. These latter structures have never been described in *C. elegans* but have been studied in other systems such as *Drosophila* and human cell lines, making this observation novel and exciting. *ssna-1* are maternal effect lethal, with embryos apparently dying from a multipolar spindle phenotype that arises beginning at the 2-4 cell division stage of the embryo. Multipolar spindles are associated with centrioles, indicating that cells have more than the expected 4 centrioles (one mother-daughter pair and one grandmother-daughter pair). This extra centriole phenotype could plausibly arise from overduplication, from premature centriole disengagement, or from centriole fragmentation. The authors find that the extra centriole phenotype observed in *ssna-1* mutants is not due to overexpression of centriole components and can't be suppressed by a mutation in *zyg-1*. Instead, a *zyg-1* mutation enhances the *ssna-1* phenotype at 20 degrees and SSNA-1 protein co-sediments with ZYG-1 protein. To distinguish between centriole fragmentation and premature disengagement, the authors perform an elegant marked mating experiment to track the fate of the paternally deposited centrioles in *ssna-1* mutants. They find that centrioles appear to fragment, which they propose is the root cause of the multipolar spindle phenotype. Since *ssna-1* appears genetically interacts with *sas-1*, the authors propose a model where SSNA-1 works with SAS-1 to promote centriole stability. Overall, the data are solid and well-presented and the study represents an important advance in our understanding of the centriole and of the function of Sjogren's Syndrome Nuclear Antigen 1. A few clarifications and additional descriptions would make the paper stronger.

Major:

1. I really like the marked mating experiment as a means to demonstrate centriole fragmentation, however, I can still come up with a complicated explanation for why this phenotype might arise from premature disengagement, especially if there is a defect in *ssna-1* sperm. The simplest explanation for the phenotypes observed in *ssna-1* mutants is centriole fragmentation, however, although it is surprising that these centriole fragments are stable enough to serve as seemingly functional spindle poles. The argument that defects in *ssna-1* mutants are due to centriole fragmentation would be bolstered by additional characterization of the fragments. This could be accomplished in many ways including:

- 1) A demonstration that older embryos have more fragments: I am surprised by how few and consistent the number of fragments is. Why does only one centriole appear to fragment, i.e. that you only ever have 3 spots (Figure 6D).
- 2) Further imaging of the fragments: This could be accomplished using the expansion protocol used in Figure 3 and would allow a more in-depth characterization of what has gone wrong in *ssna-1* centrioles. Of course, this could also be accomplished by EM, but such a labor intensive experiment is not expected within the review period.
- 3) Further molecular characterization of the fragments: The authors show that SAS-4 localizes to fragments but have not characterized them in any other way. Given the genetic interaction with *sas-1*, it would be good at very least to test if SAS-1 still localizes to fragments, but other characterization would be welcome as well.

2. Perhaps related to the above point, the authors present evidence that procentrioles are apparently generated by anaphase in *C. elegans* (as in other species such as *Chlamydomonas*): There is a small dot of green SAS-4 with the red paternal centriole in panel 1 in Fig 6C wt mating panel and there is a dot of SAS-4 with the red SSNA-1 dot in Fig. S1E. Both

of these results indicate that the old centriole provided by the sperm has a procentriole with it. It would be great to call out this result as it has been suspected and would be an important piece of information for the community

3. It is impossible to assess the claims of the companion paper Agostini et al., 2024. Thus, effects of the various *ssna-1* mutants on protein function should be presented as speculative. For example, no data provided in this paper shows that the R18 residue is required for MT binding, thus the authors can't conclude that MT binding is required for SSNA-1 function. Similarly, viability data for these mutants should be shown in this paper. Line 250 should make clear they are referencing the companion paper as the current callout is confusing.

Minor:

1. Introduction:

- Can the authors describe a bit more about what is known about Sjogren's Syndrome Nuclear Antigen 1 in a disease context?
- Centriole elimination occurs in many cell types, and it would be good to expand the description beyond just female meiosis.

2. Color nitpicks:

- I would encourage the authors to use color-blind friendly combinations of colors, e.g green/magenta instead of green/red.
- In the setup for Fig. 1E, can the mutant be one color and wt be another in Figure 1E? This case would be easier to compare.
- In Figure 7, define green and purple features in the centriole cartoon, ideally on the cartoon, but at least in the figure legend.
- The cartoon in Figure 6B is confusing because I assume wt should have two green and two yellow poles regardless of what is occurring in the mutant case, but the top case presents the wt as if one pole should be all red. I assume both poles in wt that come from a sperm derived centriole should also have a newly synthesized daughter centriole.

3. The authors write the paper early on as if centriole overduplication is the only explanation for the multipolar phenotype. I was distracted by this setup given that there are many possible reasons for the extra puncta they observe. I feel their arguments would be more effective if they set up that there are multiple sources for extra SAS-4 containing puncta and then eliminate those possibilities one by one. For example, line 302 just assumes a centriole amplification defect rather than presenting this as a possibility that is being tested.

4. Figure S1: Other contexts in which SSNA-1 localizes (e.g. sperm and head cilia) could use some cartoons to make them accessible to non *C. elegans* readers.

5. Figure S4B needs images

6. It is unclear why *ssna-1* centriole fragmentation phenotypes only emerge in the 2-4 cell division, especially given the extreme pulling forces in the one cell embryo. Can the authors speculate about this in the discussion?

Reviewer #2

(Remarks to the Author)

In this study, the authors investigate the role of a *C.elegans* SSNA1 orthologue in centriole structural integrity. SSNA1 is a microtubule-associated protein previously shown to localize to centrosomes in human cells. Using *C. Elegans* as a model, the authors report that SSNA1 localizes to centrioles close to the outer microtubule wall and centriole satellite structures. Knockout of SSNA1 using CRISPR-Cas9 results in centriole amplification due to fragmentation, as well as multipolar spindle defects. The authors investigate colocalization, genetic and physical interactions of SSNA1 with several known centriolar proteins. The authors conclude that SSNA1 is a centriole stabilizing protein acting post centriole assembly. Overall, the study characterizes an interesting centriolar component that was previously not studied in this context. The main issue is that many of the interpretations presented rely on the findings that are presumably a part of a parallel in vitro study. However, this in vitro study (reference #36, Agostini et al.) is not available. Thus, the manuscript needs to be revised and the results re-interpreted in the light of the presented data and the available literature only.

Specific comments:

102-104: 'in conjunction with a parallel in vitro study' needs revision!

121: 'our parallel study'

This section needs to be revised, given the lack of the availability of the parallel study. The referenced structural data and biochemical activity in vitro are not shown/available.

154-156 Another reference to unavailable structural study. The results of small deletions should be re-interpreted.

Fig 3:

- Panels ABC could use the same pseudocolor scheme as DE?
- The authors conclude that SSNA-1 localizes within the vicinity of SAS-1, but they don't show SAS-1? Should show SAS-1 as well!

250 Another reference to unavailable structural study

256 – 258 Relies on the results of the unavailable study

This conclusion does not stand without the accompanying study and should thus be removed from the manuscript or re-interpreted.

282 – 283 Relies on the results of the unavailable study

Fig. 5DE: How come no SSNA1 in supernatant?

Physical interaction (e.g. pulldown) between SSNA1 and SAS1 should also be investigated.

Does SSNA1 overexpression rescue SAS1 knockdown?

The authors should use a color-blind friendly color scheme in figures.

Reviewer #3

(Remarks to the Author)

Centrioles are unique stable structures within cells, but the mechanisms underlying their stability are not well-understood. This manuscript by Pfister et al identifies a microtubule binding protein, SSNA-1, as essential for centriole stability in *C. elegans*. This is an excellent study: interesting, novel, and well executed. The authors take advantage of *C. elegans* genetics and combine it with high resolution imaging to demonstrate that SSNA-1 is a component of centrioles and its loss results in multipolar spindle formation. A major highlight of the paper is the authors' ability to distinguish between defects in centriole assembly and defects post assembly by marking paternally inherited centrioles. The major points in the paper are well-supported, and this manuscript should be published with revisions addressing the following major and minor comments.

Major comments:

1. A major claim of the paper is that loss of *ssna-1* results in centriole fragmentation. The data would be stronger if authors can show what these centriole fragments look like at higher resolution by expansion microscopy or TEM. Are they fragmenting laterally such that the proximal and distal regions are separated, or are they fragmenting longitudinally? This would be important for understanding SSNA-1 function: for example, longitudinal fragmentation would suggest that SSNA-1 is required to stabilize and hold nine singlet microtubules together. These results would also define the minimal requirements for MTOC formation: is a fragmented half centriole sufficient to serve as an MTOC?
2. The authors demonstrate that SSNA-1 localizes within the lumen of centrioles in *C. elegans* gonads. However, the authors also show that SSNA-1 does not have a functional role within the gonad, and thus its localization may differ in embryos, where SSNA-1 is required for cell division and embryonic viability. The authors should localize SSNA-1 within embryos using expansion microscopy.
3. Centrioles are paternally inherited organelles, but surprisingly, the authors demonstrate that loss of paternal *ssna-1* does not result in embryonic lethality. The authors propose that cytoplasmic SSNA-1 in the egg can be recruited to centrioles. However, the authors also demonstrate that SSNA-1 does not turn over after fertilization and cell division (Fig S1E). To resolve this question, the authors should directly test whether cytoplasmic SSNA-1 can be recruited to sperm centrioles. They can cross *ssna-1* mutant males expressing tagged SAS-4 to wildtype hermaphrodites expressing tagged SSNA-1. If the hypothesis is correct, all centrioles in the embryo, including the paternally contributed tagged SAS-4 centrioles, should also have tagged SSNA-1.
4. The authors should provide genotyping information for the knockout mutants. I also did not see any rescue experiments; these should be performed.

Minor comments:

1. On p8, line 171, the authors write that “the strain expressing SSNA-1::SPOT lacks an embryonic lethal phenotype indicating the tagged protein is functional.” Are both alleles tagged? If not, it could be that the untagged SSNA-1 is compensating for the tagged protein.
2. In Fig 3E, the diagram indicates that the diameter of the SAS-4 ring is smaller than tubulin, indicating that SAS-4 lies within the centriole lumen. However, the images in Fig 3A show that SAS-4 is outside the centriole and likely has a larger diameter than tubulin. The authors should amend 3E to better reflect the data.
3. Fig 6B: stable and unstable are misspelled

Version 1:

Reviewer comments:

Reviewer #1

(Remarks to the Author)

Here, Pfister et al. submit a revised manuscript describing the role of SSNA-1 in *C. elegans* centriole biology. The authors

have addressed all of my concerns and significantly strengthened the paper. I endorse publication at this stage. I have included a few small comments below.

Line 287, 288 exhibit[ed]

Please update Movie S1 to be magenta/green color scheme as the rest of the figures.

Upon re-reading, I still am fixated on the small dot of green SAS-4 next to the magenta centriole consistent with the idea that a procentrioles are born at the end of mitosis as in other organisms like *Chlamydomonas*. Given this is occurring in wild-type mitosis, it seems plausible that the sperm is actually donating a mother and daughter centriole AND two pro-centrioles. Thus, theoretically, there could be 4 magenta dots in the marked mating experiment shown in Figure 6. This idea is very speculative, so I leave it up to the authors to comment on this possibility, but it will be a viable hypothesis in my mind in addition to fragmentation.

Reviewer #2

(Remarks to the Author)

In their revised manuscript, the authors have adequately addressed my previous concerns.

Reviewer #3

(Remarks to the Author)

This is an excellent revision and the authors have addressed all of my comments. It is unfortunate that fragmenting centrioles cannot be visualized at higher resolution, but I understand that this is a challenging experiment to perform on the relatively small *C. elegans* centrioles. I recommend this manuscript for publication in Nature Communications.

Reviewer #1

Major:

1. I really like the marked mating experiment as a means to demonstrate centriole fragmentation, however, I can still come up with a complicated explanation for why this phenotype might arise from premature disengagement, especially if there is a defect in *ssna-1* sperm. The simplest explanation for the phenotypes observed in *ssna-1* mutants is centriole fragmentation, however, although it is surprising that these centriole fragments are stable enough to serve as seemingly functional spindle poles. The argument that defects in *ssna-1* mutants are due to centriole fragmentation would be bolstered by additional characterization of the fragments. This could be accomplished in many ways including:

1) A demonstration that older embryos have more fragments: I am surprised by how few and consistent the number of fragments is. Why does only one centriole appear to fragment, i.e. that you only ever have 3 spots (Figure 6D).

We don't actually see more fragments in older embryos. The reason for this might be that some fragments lack long term stability. As for why we don't see both sperm-derived centrioles fragmenting in the same embryo, this is likely a matter of probability. In Figure 6D, we observed a total of 7 fragmented centrioles out of 70 analyzed in the *ssna-1(Δ)* strain during the first two cell cycles. Thus the probability of a single centriole fragmenting sometime during the first two cell cycles is 0.1 and the probability of both sperm-derived centrioles fragmenting during this time-period is the square of this number or 0.01. When analyzing 70 centrioles we would expect on average to observed just 0.7 cases where both sperm derived centrioles fragment in the same embryo. Another thing to consider is the small size of *C. elegans* centrioles (100 x150 nm). There might be a limit to how small a fragment can be, while maintaining its stability (as well as our ability to detected it). Thus it is possible that centrioles in the *ssna-1* mutant do fragment more often than our numbers suggest but that some of the fragments produced are too small to maintain stability or be detected.

2) Further imaging of the fragments: This could be accomplished using the expansion protocol used in Figure 3 and would allow a more in-depth characterization of what has gone wrong in *ssna-1* centrioles. Of course, this could also be accomplished by EM, but such a labor intensive experiment is not expected within the review period.

Thank you for this suggestion. We have performed expansion microscopy on *ssna-1(Δ)* embryos and have imaged multipolar spindles. Unfortunately, we were unable resolve centriole fragments. This is likely because *C. elegans* centrioles are very small, and the fragments even smaller. Thus our existing U-ExM methods simple do not have the resolution required for such analysis. We believe a thorough analysis of how centrioles break and how the absence of SSNA-1 ultimately affects centriole structure will require EM but as the reviewer points out this is labor intensive and perhaps beyond the scope of the current study. Nonetheless we feel the additional data, particularly the new data

showing a loss of SAS-1 in *ssna-1*(Δ) centrioles, strongly aligns with the results of our centriole fragmentation assay.

3) Further molecular characterization of the fragments: The authors show that SAS-4 localizes to fragments but have not characterized them in any other way. Given the genetic interaction with *sas-1*, it would be good at very least to test if SAS-1 still localizes to fragments, but other characterization would be welcome as well.

We now include several experiments aimed at further characterization of the fragments. As shown in the paper these fragments also contain SPD-2 (Fig 4A), ZYG-1 (Fig 4F) and SAS-7 (Figure 6A). However we also added additional data showing that the fragments contain other centriole proteins such as SAS-5 (Fig. S3C) SAS-6(Fig. S3D), and SAS-1 (Fig S3E). Additionally we have added new data showing that centrioles lacking SSNA-1 also possessed reduced levels of SAS-1 (Fig S5F). Since SAS-1 has an established role in centriole stability (<https://doi.org/10.1371/journal.pgen.1004777>), this later finding bolsters our contention that centrioles are fragmenting in the *ssna-1*(Δ) mutant. However we decided to go a step further and quantified the intensity of the SPOT::SAS-4 foci in our fragmentation assay (Fig 6E) and found that in mutant embryos with three foci, one was always much brighter than the other two (presumably fragments) (Fig. 6E). In contrast, in wild-type embryos the two sperm-derived centrioles were much more similar in intensity. These results are consistent with the extra centrioles arising through breakage rather than the production of extra centrioles. Taken all together our results provide very strong evidence that centrioles are fragmenting in the *ssna-1* mutant

2. Perhaps related to the above point, the authors present evidence that procentrioles are apparently generated by anaphase in *C. elegans* (as in other species such as *Chlamydomonas*): There is a small dot of green SAS-4 with the red paternal centriole in panel 1 in Fig 6C wt mating panel and there is a dot of SAS-4 with the red SSNA-1 dot in Fig. S1E. Both of these results indicate that the old centriole provided by the sperm has a procentriole with it. It would be great to call out this result as it has been suspected and would be an important piece of information for the community

We thank the reviewer for pointing this out. We now call attention to this in the legend of Fig. 6.

3. It is impossible to assess the claims of the companion paper Agostini et al., 2024. Thus, effects of the various *ssna-1* mutants on protein function should be presented as speculative. For example, no data provided in this paper shows that the R18 residue is required for MT binding, thus the authors can't conclude that

MT binding is required for SSNA-1 function. Similarly, viability data for these mutants should be shown in this paper. Line 250 should make clear they are referencing the companion paper as the current callout is confusing.

It appears that this reviewer as well as reviewer 2 did not have access to the Agostini et al. paper that was co-submitted with our manuscript. However, we did provide a copy of this paper along with our initial submission so that the reviewers could assess our claims. After receiving the reviewers' comments, we addressed this issue with the editor who told us it would be emailed to the reviewers. We hope that all reviewers are now able to access the referenced manuscript. We have reworded the reference on line 250 to improve clarity.

Minor:

1. Introduction:

- Can the authors describe a bit more about what is known about Sjogren's Syndrome Nuclear Antigen 1 in a disease context?

Sjogren's Syndrome is an autoimmune disorder that results in the production of antibodies against self-antigens. One of these self-antigens is SSNA1. Importantly mutations in the SSNA1 gene are not linked to this disease and only about 13% of SS patients produce SSNA1 antibodies (Nozawa et al. Front Biosci. 2009 Jan 1;14(10):3733-9. doi: 10.2741/3484.) indicating that the immune response to SSNA-1 is unlikely to contribute to the disease pathology. So in summary there really is nothing connecting SSNA1 to the disease other than the protein is an occasional target of SS patients' immune system.

- Centriole elimination occurs in many cell types, and it would be good to expand the description beyond just female meiosis.

This is a good point, and we have adjusted this statement in the introduction accordingly.

2. Color nitpicks:

- I would encourage the authors to use color-blind friendly combinations of colors, e.g green/magenta instead of green/red.

Thank you for this suggestion. We have converted all figures to magenta/cyan/yellow color combinations.

- In the setup for Fig. 1E, can the mutant be one color and wt be another in Figure 1E? This case would be easier to compare.

We agree. Done

- In Figure 7, define green and purple features in the centriole cartoon, ideally on the cartoon, but at least in the figure legend.

Done.

- The cartoon in Figure 6B is confusing because I assume wt should have two green and two yellow poles regardless of what is occurring in the mutant case, but the top case presents the wt as if one pole should be all red. I assume both poles in wt that come from a sperm derived centriole should also have a newly synthesized daughter centriole.

The reviewer is correct. We have adjusted the cartoon.

3. The authors write the paper early on as if centriole overduplication is the only explanation for the multipolar phenotype. I was distracted by this setup given that there are many possible reasons for the extra puncta they observe. I feel their arguments would be more effective if they set up that there are multiple sources for extra SAS-4 containing puncta and then eliminate those possibilities one by one. For example, line 302 just assumes a centriole amplification defect rather than presenting this as a possibility that is being tested.

The reviewer raises a great point and we have adjusted the text in several places to make it clear that multiple centrioles can arise from anyone of several defects (overduplication, reduplication, fragmentation). This includes the text on line 302. Also we intended to write the paper in the precise way the reviewer suggests by addressing each possibility and eliminating them one by one. While the manuscript follows this general pattern, it is not feasible to address each possibility precisely one after the other. For instance our assay for centriole fragmentation also addresses premature disengagement, so these possibilities are addressed together. Nonetheless we think the revised version of our manuscript very clearly outlines the possibilities and how they are addressed.

4. Figure S1: Other contexts in which SSNA-1 localizes (e.g. sperm and head cilia) could use some cartoons to make them accessible to non C. elegans readers.

Done

5. Figure S4B needs images

Done

6. It is unclear why ssna-1 centriole fragmentation phenotypes only emerge in the 2-4 cell division, especially given the extreme pulling forces in the one cell embryo. Can the authors speculate about this in the discussion?

One possibility to explain why centrioles fragment at the 2-4 cell stage and not earlier is that the microtubule-dependent forces are not strong enough to immediately fracture centrioles lacking SSNA-1 and can only gradually weaken them over time until they eventually succumb. This also might explain why the frequency of centriole fragmentation is low. We have modified this discussion to include this idea.

Reviewer #2 (Remarks to the Author):

The main issue is that many of the interpretations presented rely on the findings that are presumably a part of a parallel in vitro study. However, this in vitro study (reference #36, Agostini et al.) is not available. Thus, the manuscript needs to be revised and the results re-interpreted in the light of the presented data and the available literature only.

Reviewer 1 had a similar concern. As mentioned, we did provide a copy of the Agostini et al paper with our initial submission, so that the reviewers could assess our references to this work. However reviewers 1 and 2 did not seem to have access to the companion paper. Upon receiving the reviewers' comments we raised this issue with the editor who told us he would email it to the reviewers. You should therefore have access to it when reviewing our resubmitted manuscript. But if not, please contact the editor. Also please note that all major findings of our study do not rely on the work performed in the Agostini paper. The identification of *ssna-1*, its localization to the central tube of the centriole, its role in centriole stability, its genetic and physical interaction with ZYG-1 and its genetic interaction with SAS-1 are unique findings of our paper that do not rely on this other study.

Specific comments:

102-104: 'in conjunction with a parallel in vitro study' needs revision!

Editor will provide referenced manuscript.

121: 'our parallel study'

This section needs to be revised, given the lack of the availability of the parallel study. The referenced structural data and biochemical activity in vitro are not shown/available.

Editor will provide referenced manuscript.

154-156 Another reference to unavailable structural study. The results of small deletions should be re-interpreted.

Editor will provide referenced manuscript.

Fig 3:

- **Panels ABC could use the same pseudocolor scheme as DE?**

At your suggestion, we changed all images to color-blind friendly. We then altered the color scheme in Panels D and E to match panels A-C.

- **The authors conclude that SSNA-1 localizes within the vicinity of SAS-1, but they don't show SAS-1? Should show SAS-1 as well!**

Thank you for this suggestion. We have performed expansion microscopy staining for both SSNA-1 and SAS-1::3XFlag. As shown in new Figure 3F (germ line centrioles) and 3G (embryonic centrioles) we find that these two proteins nearly perfectly colocalize within the centriole.

250 Another reference to unavailable structural study

Editor will provide referenced manuscript.

256 – 258 Relies on the results of the unavailable study

This conclusion does not stand without the accompanying study and should thus be removed from the manuscript or re-interpreted.

Editor will provide referenced manuscript.

282 – 283 Relies on the results of the unavailable study

Editor will provide referenced manuscript.

Fig. 5DE: How come no SSNA1 in supernatant?

This is just the way SSNA-1 behaves. Most of the SSNA-1 in the preparation incorporates into filaments. These filaments are insoluble and therefore can be spun out of solution. As a result SSNA-1 is mostly absent from the supernatant while enriched in the pellet.

Physical interaction (e.g. pulldown) between SSNA1 and SAS1 should also be investigated.

We expressed and purified SAS-1 from *E. coli* and tested it in an in vitro pull-down assay with SSNA-1. While we did detect a weak interaction between the two proteins over background, the results were not conclusive. We plan to pursue this interaction in the future, but this will require careful optimization of the conditions for SAS-1 purification. While we were not able to accomplish this during the time frame needed for resubmission, we have included an AlphaFold model that strongly predicts an interaction between SSNA-1 and SAS-1. This model now appears in Figure 6H.

Does SSNA1 overexpression rescue SAS1 knockdown?

Thank you for suggesting this very interesting experiment. We created a codon-optimized wild-type SSNA-1 transgene whose expression was driven by the SPD-2 promoter and 3'-UTR and confirmed by IF that the protein was overexpressed. We then crossed it into a *sas-1* mutant. We found that overexpressing SSNA-1 did not decrease the embryonic lethality of a *sas-1* mutant. In fact it significantly increased the lethality of the *sas-1* mutant. This result suggests that at a molecular level, SAS-1 and SSNA-1 have distinct roles and one can't substitute for the other. As the result section is already quite long and this result turned out to be a negative one, we thought it best not to include it in the paper.

The authors should use a color-blind friendly color scheme in figures.

As suggested, we have changed the color scheme in all figures to CMYK.

Reviewer #3 (Remarks to the Author):

1. A major claim of the paper is that loss of *ssna-1* results in centriole fragmentation. The data would be stronger if authors can show what these centriole fragments look like at higher resolution by expansion microscopy or TEM. Are they fragmenting laterally such that the proximal and distal regions are separated, or are they fragmenting longitudinally? This would be important for understanding SSNA-1 function: for example, longitudinal fragmentation would suggest that SSNA-1 is required to stabilize and hold nine singlet microtubules together. These results would also define the minimal requirements for MTOC formation: is a fragmented half centriole sufficient to serve as an MTOC?

We considered using TEM but due to the low frequency of fragment formation and the need for serial sectioning, it would take a massive labor-intensive effort to achieve this in the short timeframe for resubmission. However, we attempted expansion microscopy using a relatively recently developed protocol for embryos ([10.17912/micropub.biology.001033](https://doi.org/10.17912/micropub.biology.001033)). Unfortunately, we were not able to clearly visualize centriole fragments. The likely reason for this is that *C. elegans* centrioles are small and fragments would be even smaller. The U-ExM technique did not produce the necessary level of resolution to detect and characterize the fragments. In the future we hope to further refine our U-ExM technique for embryos to better characterize these fragments, or alternatively attempt TEM.

2. The authors demonstrate that SSNA-1 localizes within the lumen of centrioles in *C. elegans* gonads. However, the authors also show that SSNA-1 does not have a functional role within the gonad, and thus its localization may differ in embryos,

where SSNA-1 is required for cell division and embryonic viability. The authors should localize SSNA-1 within embryos using expansion microscopy.

Thank you for pointing this out. At the time we performed our expansion microscopy, a method for applying this to embryos (which have a thick impermeable eggshell) had not been worked out. However members of our team recently developed such a technique ([10.17912/micropub.biology.001033](https://doi.org/10.17912/micropub.biology.001033)). We therefore co-localized SSNA-1 and SAS-1 in both gonad and embryonic centrioles and find that in both cases the two proteins closely overlap. This new data can be found in Fig 3G and F.

3. Centrioles are paternally inherited organelles, but surprisingly, the authors demonstrate that loss of paternal *ssna-1* does not result in embryonic lethality. The authors propose that cytoplasmic SSNA-1 in the egg can be recruited to centrioles. However, the authors also demonstrate that SSNA-1 does not turn over after fertilization and cell division (Fig S1E). To resolve this question, the authors should directly test whether cytoplasmic SSNA-1 can be recruited to sperm centrioles. They can cross *ssna-1* mutant males expressing tagged SAS-4 to wildtype hermaphrodites expressing tagged SSNA-1. If the hypothesis is correct, all centrioles in the embryo, including the paternally contributed tagged SAS-4 centrioles, should also have tagged SSNA-1.

As suggested, we mated *ssna-1* mutant males with hermaphrodites expressing SPOT::SSNA-1. We found that SSNA-1 is recruited to the sperm centrioles immediately after fertilization (new data in Fig. S5E). As SSNA-1 from the zygotic cytoplasm associated with centrioles during meiosis before new daughter centriole assembly is initiated (Pelletier et al. 2006. doi: 10.1038/nature05318), the newly recruited protein must be associating with the sperm-derived centrioles. Thus as predicted, SSNA-1 can assemble into centrioles post assembly. Once incorporated, it does not turn over.

4. The authors should provide genotyping information for the knockout mutants. I also did not see any rescue experiments; these should be performed.

Thank you for raising this issue. Sequencing data for the knockout mutant will be submitted along with other raw data to the figshare repository. As for the rescue experiment, we constructed a SSNA-1 transgene and crossed the transgene into the *ssna-1* deletion mutant. As shown in revised Fig. 1F, the transgene rescued the embryonic lethal phenotype completely.

Minor comments:

1. On p8, line 171, the authors write that “the strain expressing SSNA-1::SPOT

lacks an embryonic lethal phenotype indicating the tagged protein is functional.” Are both alleles tagged? If not, it could be that the untagged SSNA-1 is compensating for the tagged protein.

Yes, the strain is homozygous for the tagged version of SSNA-1.

2. In Fig 3E, the diagram indicates that the diameter of the SAS-4 ring is smaller than tubulin, indicating that SAS-4 lies within the centriole lumen. However, the images in Fig 3A show that SAS-4 is outside the centriole and likely has a larger diameter than tubulin. The authors should amend 3E to better reflect the data.

Figure 3E actually shows that tubulin and HA-SAS-4 are coincident. If one looks closely at this panel, they will see that we placed the ring of HA::SAS-4 on top of the ring on tubulin. Given that the average diameter of the tubulin ring is 98 nm and the average diameter of the HA-SAS-4 ring is 97 nm, they essentially occupy the same space.

3. Fig 6B: stable and unstable are misspelled

Thanks. Fixed.